Research





**Keywords:**
Bipedal locomotion, human running, step-down perturbation, postural stability, TSLIP model, virtual point (VP, VPP)

**Author for correspondence:**
Özge Drama
e-mail: drama@is.mpg.de

# Postural stability in human running with step-down perturbations: an experimental and numerical study

Özge Drama[1,†], Johanna Vielemeyer[2,3,†], Alexander Badri-Spröwitz[1] and Roy Müller[2,3]

[1]Dynamic Locomotion Group, Max Planck Institute for Intelligent Systems, Stuttgart, Germany
[2]Department of Neurology/Orthopedic Surgery, Klinikum Bayreuth GmbH, Germany
[3]Department of Motion Science, Friedrich Schiller University-Jena, Jena, Germany

ÖD, 0000-0001-7752-0950; RM, 0000-0002-4688-1515

Postural stability is one of the most crucial elements in bipedal locomotion. Bipeds are dynamically unstable and need to maintain their trunk upright against the rotations induced by the ground reaction forces (GRFs), especially when running. Gait studies report that the GRF vectors focus around a virtual point above the centre of mass ($VP_A$), while the trunk moves forward in pitch axis during the stance phase of human running. However, a recent simulation study suggests that a virtual point below the centre of mass ($VP_B$) might be present in human running, because a $VP_A$ yields backward trunk rotation during the stance phase. In this work, we perform a gait analysis to investigate the existence and location of the VP in human running at $5\,\mathrm{m\,s^{-1}}$, and support our findings numerically using the spring-loaded inverted pendulum model with a trunk. We extend our analysis to include perturbations in terrain height (visible and camouflaged), and investigate the response of the VP mechanism to step-down perturbations both experimentally and numerically. Our experimental results show that the human running gait displays a $VP_B$ of ≈−30 cm and a forward trunk motion during the stance phase. The camouflaged step-down perturbations affect the location of the $VP_B$. Our simulation results suggest that the $VP_B$ is able to encounter the step-down perturbations and bring the system back to its initial equilibrium state.

## 1. Introduction

Bipedal locomotion in humans poses challenges for stabilizing the upright body owing to the under-actuation of the trunk and the hybrid dynamics of the bipedal structure (table 1).

[†]Shared first authorship.


**Table 1.** Nomenclature.

| | |
|---|---|
| general terminology | |
| CoM | centre of mass |
| TSLIP | spring loaded inverted pendulum model extended with a trunk |
| VP | virtual point |
| $VP_A$ | virtual point above the centre of mass |
| $VP_B$ | virtual point below the centre of mass |
| $VP_{BL}$ | virtual point below the centre of mass and below the leg axis at touch-down |
| $g$ | $g = 9.81$ m s$^{-2}$, standard acceleration due to gravity |
| symbols related to the experiment | |
| $l$ | distance between lateral malleolus and trochanter major of the leg in contact with the ground |
| CoP | centre of pressure |
| GRFs | ground reaction forces |
| V0 | experiment with level ground |
| V10 | experiment with 10 cm visible step-down perturbation |
| C10 | experiment with 10 cm camouflaged step-down perturbation |
| $R^2$ | coefficient of determination |
| $\gamma$ | the trunk angle estimated from markers on L5 and C7. The trunk angle $\gamma$ corresponds to the $\theta_C$ in the TSLIP model |
| $N_{trial}$ | number of trials |
| $N_{\%}$ | number of gait percentage times analysed |
| $\theta_{exp}$ | angle of the experimental measured GRFs |
| $\overline{\theta}_{exp}$ | mean experimental angle of GRFs |
| $\theta_{theo}$ | angle of theoretical forces |
| $\vec{\boldsymbol{p}}$ | impulse |
| $\vec{\boldsymbol{p}}_{normalized}$ | normalized impulse |
| $\vec{\boldsymbol{p}}_{brake}$ | braking impulse |
| $\vec{\boldsymbol{p}}_{prop}$ | propulsion impulse |
| symbols related to the simulations | |
| $[x_C, z_C, \theta_C]$ | state vector of the centre of mass |
| $[r_{FC}, r_{FV}, r_{FH}]$ | position vectors from foot to the centre of mass, virtual point and hip joint, respectively |
| $\Delta z$ | step-down height |
| $m$ | mass |
| $J$ | moment of inertia |
| $l$ | leg length |
| $\theta_L$ | leg angle |
| $\tau_H$ | hip torque |
| $F_{sp}$ | leg spring force |
| $F_{dp}$ | leg damper force |
| ${}_F\mathbf{F}_a$ | axial component of the ground reaction force in foot frame |
| ${}_F\mathbf{F}_t$ | tangential component of the ground reaction force in foot frame |
| $r_{VP}$ | VP radius, the distance between the centre of mass and virtual point |
| $\theta_{VP}$ | VP angle, the angle between trunk axis and $VP_A$, or the vertical axis passing from CoM and $VP_B$ |
| superscripts | |
| AP | apex event, where the centre of mass reaches to its maximum height |

| | |
|---|---|
| TD | leg touch-down event |
| TO | leg take-off event |
| Des | desired value of the variable |
| subscripts | |
| $i$ | current step |
| $i-1$ | previous step |

Human gait studies investigate the underlying mechanisms to achieve and maintain the postural stability in symmetrical gaits such as walking and running. One major observation states that the ground reaction forces (GRFs) intersect near a virtual point (VP) above the centre of mass (CoM) [1]. Subsequent gait studies report that the VP is 15–50 cm above the CoM ($VP_A$) in the sagittal plane for level walking [1–5]. Among those, only a single study reports a limited set of level walking trials with a VP below the CoM ($VP_B$) [1]. The $VP_A$ strategy is also observed when coping with the step-down perturbations in human walking, even when walking down a camouflaged curb [5]. A similar behaviour is observed for the avians, where a $VP_A$ of 5 cm is reported for level walking, grounded running, and running of the quail [6,7]. Unlike in the studies with healthy subjects, it is reported that humans with Parkinson's disease display a $VP_B$ when walking [8]. In addition, a $VP_B$ was identified in the frontal plane for human level walking [9]. The existing literature for human running report a $VP_A$ [7,10]. However, these experiments are limited to a small subset of subjects and trials, hence are not conclusive.

The observation of the GRFs intersecting at a VP suggests that there is potentially a control mechanism to regulate the whole-body angular momentum [1,11,12]. Based on this premise, the behaviour of a VP-based postural mechanism would depend on the location and adjustment of the VP. It also raises the question whether the VP position depends on the gait type, locomotor task (e.g. control intent) and terrain conditions.

The spring loaded inverted pendulum model (SLIP) is extensively used in gait analysis owing to its capability to reproduce the key features of bipedal locomotion. The SLIP model is able to reproduce the CoM dynamics observed in human walking [13] and running [14–16]. This model can be extended with a rigid body (TSLIP) to incorporate the inertial effects of an under-actuated trunk, where the trunk is stabilized through a torque applied at the hip [1,10,12].

Based on the experimental observations, the VP is proposed as a control method to determine the hip torque in the TSLIP model to achieve postural stability [12]. The VP as a control mechanism in the TSLIP model has been implemented for human walking [4,17–20], hopping [21,22], running [12,23,24] and avian gaits [6,25]. It is also implemented and tested on the ATRIAS robot for a walking gait [26]. However, the currently deployed robotic studies are limited to a small set of gait properties (e.g. forward speed) and simple level terrain conditions.

In the simulation model, the selection of the VP position influences the energetics of the system by distributing the work performed by the leg and the hip [23,25]. A $VP_B$ in the human TSLIP model reduces the leg loading at the cost of increased peak hip torques for steady-state gaits. A $VP_A$ yields lower duty factors and hence higher peak vertical GRF magnitudes, whereas a $VP_B$ yields larger peak horizontal GRF magnitudes. Consequently, a $VP_A$ can be used to reduce the kinetic energy fluctuations of the CoM, and a $VP_B$ to reduce the potential energy fluctuations.

In human gait, the trunk moves forward during the single stance phase of walking and running, which is reversed by a backward trunk motion in the double stance phase of walking [27] and flight phase of running [10,27]. In TSLIP model simulations of human running, the trunk moves forward during the stance phase if a $VP_B$ is used, whereas it moves backward for a $VP_A$ [12,23,25,28].

One potential reason for the differences between the human and the model may be that the TSLIP model does not distinguish between the trunk and whole-body dynamics. In human walking, the trunk pitching motion is reported to be 180° out-of-phase with the whole body [2]. A $VP_A$ in the TSLIP model predicts the whole-body dynamics with backward rotation, and it follows that the trunk rotation is in the opposite direction (i.e. forward). The phase relationship between the trunk and whole-body rotation has not been published for human running, to our knowledge. However, we can indirectly deduce this relationship from the pitch angular momentum patterns. In human running, the pitch angular momentum of the trunk and the whole body are inphase, and they both become negative during stance phase, i.e. clockwise rotation of the runner [11]. The negative angular

momentum indicates that the GRFs should pass below the CoM. Therefore, a $VP_B$ in the TSLIP model can predict the whole-body dynamics with forward rotation, and the trunk rotation is in the same direction (i.e. forward).

The VP can also be used to manoeuver, when the VP target is placed out of the trunk axis [12,21]. A simulation study proposes to shift the VP position horizontally as a mechanism to handle stairs and slopes [29]. The gait analyses provide insights into the responses of GRFs to changes in terrain. In human running, step-down perturbations increase the magnitude of the peak vertical GRF. The increase gets even higher if the drop is camouflaged [30]. However, there is no formalism to describe how the VP position relates to the increase in GRFs in handing varying terrain conditions.

In the first part of our work, we perform an experimental analysis to acquire trunk motion patterns and ground reaction force characteristics during human running. Our gait analysis involves human-level running, and running over visible and camouflaged step-down perturbations of −10 cm. We expect to see a VP below the CoM ($VP_B$) shaped by the ground reaction forces, based on the results in [27], and a net forward trunk pitch motion during the stance phase, based on previous results from level running [27]. If the mechanism leading to a $VP_B$ in level running remains active, it should also extend to camouflaged, step-down perturbations. Consequently, we hypothesize to observe a $VP_B$ also in the step-down experiments.

In the second part, we perform a simulation analysis using the TSLIP model with the gait parameters estimated from our experiments. We generate an initial set of gaits that match to the experimental set-up, and extend our analysis to larger set of step-down perturbations up to −40 cm, which is close to the maximum achievable perturbation magnitude in avians [31]. We investigate whether a $VP_B$ controller is able to stabilize the gait against the step-down perturbations, and if so, how does it contribute to the energy flow in counteracting the perturbation.

# 2. Methods

## 2.1. Experimental methods

In this section, we describe the experimental set-up and measurement methods. In our experiments, 10 physically active volunteers (nine male, one female, mean ± s.d., age: 24.1 ± 3.4 years, mass: 73.8 ± 7.3 kg, height: 179.9 ± 7.6 cm) are instructed to run over a 17 m track. Prior to participation, an informed consent form was obtained from each volunteer. The experiment was approved by the local ethics committee and was in accordance to the Declaration of Helsinki. The running track has two consecutive force plates in its centre, where the first plate is fixed at ground-level, and the second one is height adjustable. We designed three sets of experiments, where the subjects were asked to run at their self-selected velocity[1] (4.9 ± 0.5 m s$^{-1}$, table 2). In the first experiment, the subjects were asked to run on a track with an even ground (V0). In the second experiment, the second force plate was lowered −10 cm, which was visible to the subjects (V10). In the third experiment, the second force plate was lowered −10 cm, and an opaque sheet was added on top of the plate on ground level to camouflage the drop. A wooden block was randomly placed between the second force plate and the opaque sheet during the course of the experiment without subject's knowledge. In other words, the subjects were not aware whether the step would be on the ground level (C0), or would be a step-down drop (C10). The step corresponding onto the first force plate is referred to as step −1, and the step to the second force plate as step 0.

All trials were recorded with eight cameras by a three-dimensional motion capture system working with infrared light. In summary, 12 spherical reflective joint markers (19 mm diameter) were placed on the tip of the fifth toe [A], malleolus lateralis [B], epicondylus lateralis femoris [C], trochanter major [D], and acromion [E] on both sides of the body as well as on L5 [F] and C7 [G] processus spinosus (figure 1). The CoM was determined with a body segment parameter method according to Winter [32]. The trunk angle $\gamma$ was calculated from the line joining C7 to L5 with respect to the vertical [33].

Further information concerning the participants, and the technical details of the measurement equipment (i.e. force plates, cameras) can be found in Müller *et al.* [30] and partly in Ernst *et al.* [34].

The method for analysing the gait data and estimating a potential VP is analogous to the gait analysis carried out for the human walking in [5]. Here, we denote the intersection point of the GRF vectors as a VP without implications for this point being above or below the CoM. To compute the VP, we use the

[1]The velocity was calculated for the stance phases of both contacts.

**Table 2.** Statistical analysis of VP, $R^2$, impulse and gait properties. (V0, visible level running; V10 visible drop of −10 cm; C10, camouflaged drop of −10 cm; VP, horizontal (*x*) and vertical (*z*) positions of the virtual point relative to the centre of mass for the 90% and the 100% dataset; $R^2$, coefficient of determination of the angles between measured ground reaction forces and forces through centre of pressure and VP; $\vec{p}_{brake}$, braking impulse and $\vec{p}_{prop}$, propulsion impulse in the *x*- and *z*-direction. Data are means ± s.d. across all included subjects ($n = 10$; exception: duty factor is only calculated for nine subjects) for step −1 (pre-perturbed contact) and step 0 (perturbed contact). *Post hoc* analysis with Šidák correction revealed significant differences between ground conditions: differences from V0 and V10 are indicated with 'a' and 'b', respectively ($p < 0.05$).)

| | | V0 | V10 | C10 | *p*-value | *F*-value/$\eta^2$ |
|---|---|---|---|---|---|---|
| step −1 | **VP variables** | | | | | |
| | $VPx_{100\%}$ [cm] | −2.9 ± 2.9 | −8.5 ± 3.5[a] | −8.6 ± 3.1[a] | **0.000** | 224.38/0.01 |
| | $VPx_{90\%}$ [cm] | −3.4 ± 2.8 | −8.7 ± 3.4[a] | −9.1 ± 3.2[a] | **0.000** | 146.41/0.01 |
| | $VPz_{100\%}$ [cm] | −31.5 ± 4.9 | −31.3 ± 5.0 | −31.7 ± 6.6 | 0.965 | 0.04/0.00 |
| | $VPz_{90\%}$ [cm] | −30.8 ± 5.8 | −30.7 ± 5.2 | −31.5 ± 6.5 | 0.997 | 0.23/0.00 |
| | $R^2_{100\%}$ [%] | 76.0 ± 14.6 | 79.0 ± 12.1 | 77.3 ± 13.2 | 0.424 | 0.90/0.00 |
| | $R^2_{90\%}$ [%] | 88.1 ± 3.4 | 89.4 ± 3.4 | 88.5 ± 3.1 | 0.411 | 1.45/0.00 |
| | **impulse** | | | | | |
| | $\vec{p}_{brake,x}$ | −0.05 ± 0.02 | −0.05 ± 0.02 | −0.04 ± 0.02 | 0.162 | 2.02/0.00 |
| | $\vec{p}_{brake,z}$ | 0.53 ± 0.11 | 0.47 ± 0.10 | 0.49 ± 0.06 | 0.051 | 3.53/0.01 |
| | $\vec{p}_{prop,x}$ | 0.11 ± 0.01 | 0.12 ± 0.02 | 0.11 ± 0.01 | 0.078 | 2.94/0.00 |
| | $\vec{p}_{prop,z}$ | 0.56 ± 0.02 | 0.57 ± 0.04 | 0.55 ± 0.04 | 0.421 | 0.91/0.00 |
| step 0 | **VP variables** | | | | | |
| | $VPx_{100\%}$ [cm] | −2.8 ± 4.5 | −4.0 ± 4.6[a] | −7.1 ± 5.1[a] | **0.014** | 7.95/0.01 |
| | $VPx_{90\%}$ [cm] | −2.6 ± 4.6 | −4.3 ± 4.7 | −7.0 ± 5.0[a] | **0.018** | 7.17/0.01 |
| | $VPz_{100\%}$ [cm] | −35.2 ± 6.1 | −38.8 ± 5.6[a] | −24.6 ± 14.5 | **0.047** | 5.17/0.10 |
| | $VPz_{90\%}$ [cm] | −35.0 ± 6.3 | −37.6 ± 5.7 | −24.0 ± 16.4 | 0.074 | 4.04/0.10 |
| | $R^2_{100\%}$ [%] | 81.9 ± 11.3 | 64.1 ± 15.9[a] | 65.1 ± 13.4 | **0.021** | 6.87/0.17 |
| | $R^2_{90\%}$ [%] | 92.0 ± 2.1 | 83.0 ± 5.9[a] | 69.4 ± 8.7[a,b] | **0.000** | 70.13/0.13 |
| | **impulse** | | | | | |
| | $\boldsymbol{p}_{brake,x}$ | −0.10 ± 0.02 | −0.11 ± 0.03 | −0.04 ± 0.02[a,b] | **0.000** | 40.27/0.01 |
| | $\boldsymbol{p}_{brake,z}$ | 0.69 ± 0.08 | 0.83 ± 0.12[a] | 0.63 ± 0.12[b] | **0.000** | 20.92/0.10 |
| | $\boldsymbol{p}_{prop,x}$ | 0.09 ± 0.02 | 0.09 ± 0.01 | 0.06 ± 0.01[a,b] | **0.000** | 14.26/0.00 |
| | $\boldsymbol{p}_{prop,z}$ | 0.46 ± 0.08 | 0.48 ± 0.05 | 0.45 ± 0.06 | 0.309 | 1.19/0.01 |
| | **gait properties** | | | | | |
| | velocity [m $s^{-1}$] | 4.9 ± 0.5 | 4.9 ± 0.5 | 5.1 ± 0.4 | 0.148 | 2.13/0.11 |
| | stance time [s] | 0.18 ± 0.02 | 0.17 ± 0.02[a] | 0.14 ± 0.01[a,b] | **0.000** | 62.67/0.00 |
| | duty factor [%] | 26.7 ± 2.0 | 24.8 ± 1.6[a] | 22.4 ± 1.5[a,b] | **0.008** | 37.20/0.01 |

instantaneous GRF vectors, which have an origin at the centre of pressure (CoP) and are expressed in a CoM-centred coordinate frame that aligns with the gravity vector in the vertical axis [3]. The CoP is calculated from the kinetic data using the method described in Winter [32]. Then, the VP is estimated as the point, which minimizes the sum of the squared distances between the GRF vectors and itself. For the camouflaged setting with a wooden block placed on the force plate (C0), we can not calculate the CoP accurately. Thus, the VP is not estimated for the C0 case.

The human gait data involves impact forces at the leg touch-down, which introduces an additional behaviour in the GRF pattern [30,35,36]. In order to see the influence of the impact on VP, we are presenting our recorded data in two ways. The first calculation involves the full GRF data from leg touch-down to take-off (100% dataset), whereas the second calculation involves the GRF data starting from 10% of the stance to the leg take-off (90% dataset).

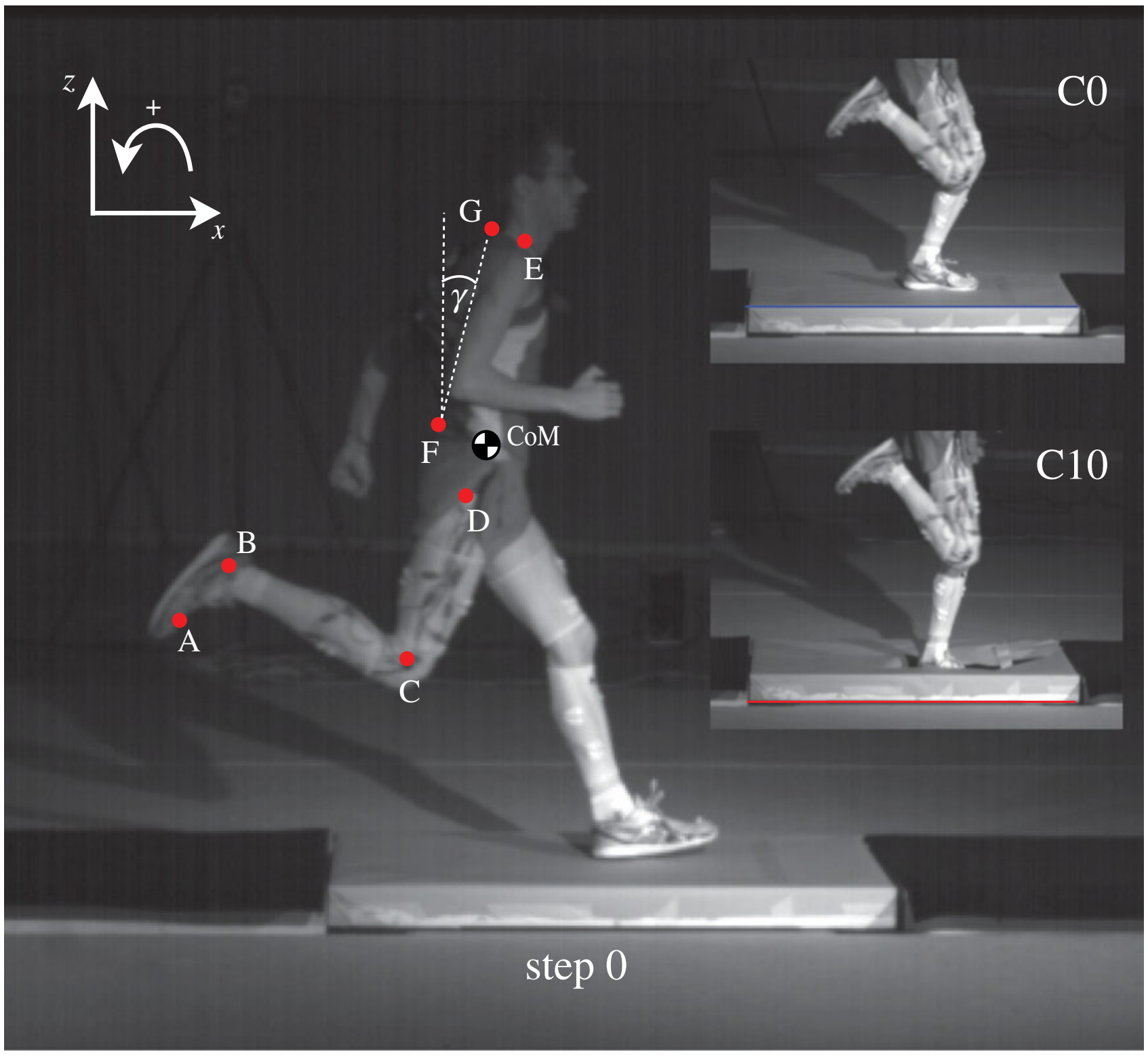


**Figure 1.** Experimental set-up. The first force plate is on the ground level, whereas the second force plate is height adjustable (step 0). The camouflaged setting for the second force plate is shown on the right for elevations of 0 cm (C0, blue) and − 10 cm (C10, red). The placement of the motion capture markers is given on the left, where the markers are denoted the letters A–G. The trunk angle is shown with $\gamma$ and is positive in the counterclockwise direction.

In the VP concept, all of the GRF vectors start from the CoP and point to a single VP. However, the human gait data differs from this theoretical case, as the human is more complex. To evaluate the amount of agreement between the theoretical VP-based forces and experimentally measured GRFs, we use a measure called the *coefficient of determination* ($R^2$) similar to Herr & Popovic [37]:

$$R^2 = \left(1 - \frac{\sum_{i=1}^{N_{\text{trial}}} \sum_{j=1}^{N_{\%}} \left(\theta_{\text{exp}}^{ij} - \theta_{\text{theo}}^{ij}\right)^2}{\sum_{i=1}^{N_{\text{trial}}} \sum_{j=1}^{N_{\%}} \left(\theta_{\text{exp}}^{ij} - \overline{\theta}_{\text{exp}}\right)^2}\right) \times 100\%. \tag{2.1}$$

The ($\theta_{\text{exp}}$, $\theta_{\text{theo}}$) are the experimental GRF and theoretical force vector angles, $N_{\text{trial}}$ is the number of trials, and $N_{\%} = 100$ is the measurement time. Here, $\overline{\theta}_{\text{exp}}$ is the grand mean of the experimental GRF angles over all trials and measurement times. The number of trials is equal to 30 for visible conditions (15 for V0 and 15 for V10) and 20 for the camouflaged conditions (12 for C0 and 8 for C10).

Note that $R^2 = 100\%$ if there is a perfect fit for the experimental GRF and the theoretical force vector angles. The value of $R^2$ approaches zero as the estimation of the model is equal to the use of $\theta_{\text{exp}}$ as an estimator [37].

We also compute the horizontal and vertical impulses $\vec{p}$ for two intervals (braking and propulsion) by integrating the GRFs over time. The braking interval went from touch-down to mid-stance (zero-crossing of the horizontal GRFs) and the propulsion interval mid-stance onwards. We report the values for brake-propulsion intervals individually in §3.1. To enable the comparison among subjects, we normalize the impulses to each subject's body weight (BW), leg length ($l$, the distance between lateral malleolus and trochanter major of the leg in contact with the ground) and standard gravity ($g$) in accordance with [38] as,

$$\vec{p}_{\text{normalized}} = \frac{p}{\text{BW} \cdot \sqrt{l/g}}. \tag{2.2}$$

Because of the inaccuracy in calculating the CoP, we did not analyse the C0 statistically. For all other experimental settings (V0, V10 and C10), we used repeated measures ANOVA ($p < 0.05$) with post hoc

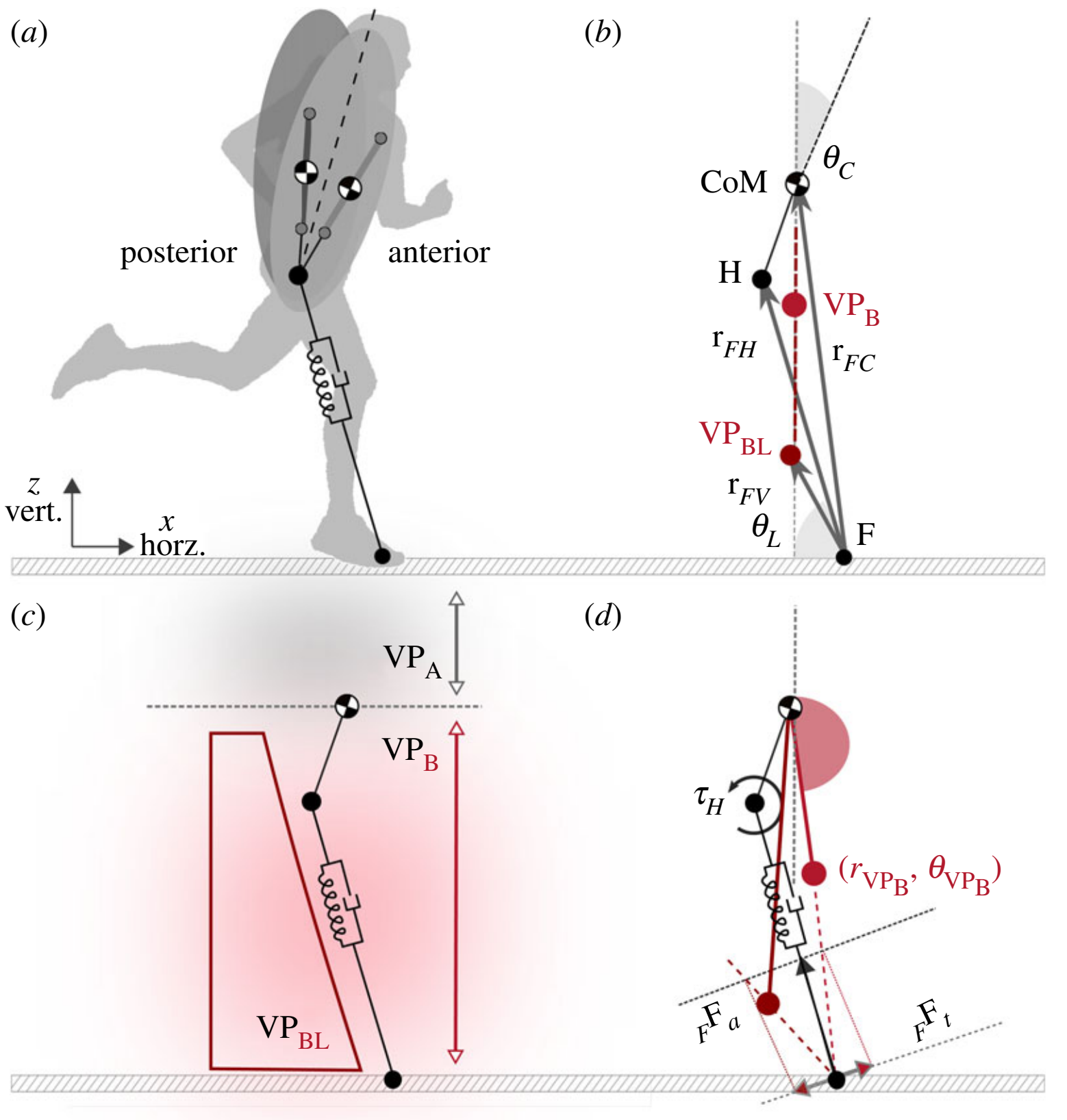


**Figure 2.** (*a*) TSLIP model that shows the forward (anterior) and backward (posterior) trunk motion. (*b*) Vector notations used in equations of motion. (*c*) The parameter space for the VP is divided into two regions: the virtual points above the centre of mass ($VP_A$) and below ($VP_B$). $VP_A$ causes backward and $VP_B$ causes forward trunk rotation during the stance phase. Each subspace is divided further with respect to the leg axis, where the sign of the hip torque changes. (*d*) For $VP_B$, the points above the leg axis yield a negative and points below ($VP_{BL}$) yield a positive hip torque at touch-down. The VP is described with the radius ($r_{VP}$) and angle ($\theta_{VP}$) that is expressed in the CoM centred world coordinate frame. Here, presented human running experiments reveal that the VP is −30 cm below the CoM (see §3.1). This corresponds to the $VP_{BL}$ region with −180° VP angle in our simulation.

analysis (Šidák correction) to test the statistical significance of the estimated VP position, the impulses and additional gait properties. In order to verify whether the VP is above or below the CoM ($VP_A$ or $VP_B$), we performed a one-sample *t*-test compared with zero, separately for each condition with Šidák correction as the *post hoc* test.

## 2.2. Simulation methods

In this section, we describe the TSLIP model that we use to analyse how the VP reacts to the step-down perturbations in human running. The TSLIP model consists of a trunk with mass $m$ and moment of inertia $J$, which is attached to a massless leg of length $l$ and a massless point foot $F$ (figure 2*a*). The leg is passively compliant with a parallel spring-damper mechanism, whereas the hip is actuated with a torque $\tau_H$. The dynamics of the system is hybrid, which involves a flight phase that has ballistic motion, followed by a stance phase that reflects the dynamics of the spring-damper-hip mechanism. The phases switch when the foot comes in contact with the ground at touch-down, and when the leg extends to its rest length $l_0$ at take-off.

The equations of motion for the CoM state ($x_C$, $z_C$, $\theta_C$) during the stance phase can be written as in equation (2.3), where the linear leg spring force $F_{sp} = k\,(l - l_0)$ and bilinear leg damping force $F_{dp} = c\,\dot{l}\,(l - l_0)$ generate the axial component of the GRF in foot frame $_F\mathbf{F}_a = (F_{sp} - F_{dp})[-\cos\theta_L \;\sin\theta_L]^{\mathrm{T}}$. Here, $k$ refers to the spring stiffness and $c$ to the damping coefficient. The hip torque $\tau_H$ creates the tangential component of the GRF $_F\mathbf{F}_t = (-\tau_H/l_L)[\sin\theta_L \;-\cos\theta_L]^{\mathrm{T}}$ (figure 2*d*):

$$m\begin{bmatrix}\ddot{x}_C\\ \ddot{z}_C\end{bmatrix} = {}_F\mathbf{F}_a + {}_F\mathbf{F}_t + g, \qquad J\,\ddot{\theta}_C = -\mathbf{r}_{FC} \times ({}_F\mathbf{F}_a + {}_F\mathbf{F}_t). \tag{2.3}$$

The leg and the hip maintain the energy balance of the system. The hip increases the system energy to propel the body forward, whereas the leg damper removes an equivalent energy in return. We determine $\tau_H$, such that the GRF points to a VP, which is characterized by the radius $r_{\rm VP}$ (i.e. distance between the hip and CoM) and angle $\theta_{\rm VP}$, as shown in figure 2*d* (red circle). The hip torque as a function of the VP is written as,

$$\begin{aligned} \tau_H = \tau_{\rm VP} &= {}_F\mathbf{F}_a \times \left[\frac{\mathbf{r}_{\rm FV} \times \mathbf{r}_{\rm FH}}{\mathbf{r}_{\rm FV} \cdot \mathbf{r}_{\rm FH}}\right] \times l, \\ \mathbf{r}_{\rm FV} &= \mathbf{r}_{\rm FC} + r_{\rm VP}\begin{bmatrix} -\sin(\theta_C + \theta_{\rm VP}) \\ \cos(\theta_C + \theta_{\rm VP}) \end{bmatrix}. \end{aligned} \tag{2.4}$$

We use two linear controllers: one for the leg angle at touch-down $\theta_L^{TD}$, and the other for the VP angle $\theta_{\rm VP}$, both of which are executed at the beginning of the step at apex, as shown in appendix A.2., figure 12. The leg angle is regulated as,

$$\theta_L^{TD}|_i = \theta_L^{TD}|_{i-1} + k_{\dot{x}_0}(\Delta \dot{x}_C^{AP}|_{-1}^{i}) + k_{\dot{x}}(\Delta \dot{x}_C^{AP}|_{i-1}^{i}), \tag{2.5}$$

with $\Delta\dot{x}|_{-1}^{i}$ being the difference in apex velocity $\dot{x}$ between time steps -1 and $i$. The VP angle is defined with respect to a CoM-centred, stationary coordinate frame that is aligned with the global vertical axis, if the VP is set below the CoM (figure 2*b*,*d*) [25]. It is adjusted based on the difference between the desired mean body angle $\theta_C^{\rm Des}$, and the mean body angle observed in the last step $\Delta\theta_C$ as,

$$\theta_{\rm VP}|_i = \theta_{\rm VP}|_{i-1} + k_{\rm VP}(\theta_C^{\rm Des} - \Delta\theta_C). \tag{2.6}$$

The model parameters are selected to match a 80 kg human with 1 m leg length (see appendix A.1., table 4 for details). The damping coefficient is set to $c = 680\,\mathrm{kNsm^{-1}}$ to match the trunk angular excursion of 4.5° reported in [27,39,40]. The forward speed and VP radius are set to $5\,\mathrm{m\,s^{-1}}$ and −30 cm, respectively, to match our estimated gait data in table 2. A VP radius of −30 cm becomes below the leg axis at leg touch-down with the model parameters we chose. Because the position of VP relative to the leg axis affects the sign of the hip torque, the $\mathrm{VP_B}$ region is separated into two and the points below the leg axis are called $\mathrm{VP_{BL}}$ (figure 2*c*,*d*), in accordance with [23].

First, we generate a base gait for level running using the framework in [23], which corresponds to the V0 in our human running experiments. Then, we introduce step-down perturbations of $\Delta z = [-10, -20, -30, -40\,\mathrm{cm}]$ in step 0. The −10 cm drop corresponds to the V10 and C10 of the human running experiments. In the simulations, the VP controller is blind to the changes in step 0, because the controller update happens only at the apex of each step. During step 0, the state of the CoM diverges from the equilibrium conditions. The postural correction starts at step 1, as the leg touch-down angle and VP angle are adjusted in response to the changes in the CoM apex state. By contrast, small adaptations might already be active at step 0 in the human experiments, e.g. resulting from swing leg retraction dynamics [16,30].

The step-down perturbation increases the total energy of the system. The added energy can be either dissipated e.g. via the hip torque or leg damper, or converted to other forms of energy e.g. change in speed or hopping height. In the latter case, we need to update the desired forward speed in the leg angle control (equation (2.5)) until all excess energy is converted to kinetic energy.

We implemented the TSLIP model in MATLAB® using variable step solver ode113 with a relative and absolute integrator error tolerance of $1 \times 10^{-12}$.

# 3. Results

## 3.1. Experimental results

The results and statistical values of the experiments are listed in table 2 and are illustrated in figures 3–5, and connected with simulation results, in figures 9–11. Additionally, significant mean differences will be highlighted in the following.

In figure 3, exemplary illustrations of the VP for single trials (V0 and C10) of different subjects at step 0 are shown. Here, the GRF vectors are plotted in a CoM-centred coordinate frame were the vertical axis is parallel to gravity. The VP is calculated as the point which minimizes the sum of squared perpendicular distances to the GRFs for each measurement time point. To avoid biases caused by the impact peak, the VP was additionally calculated for only 90% of the dataset. That means that the GRFs of the first 10%

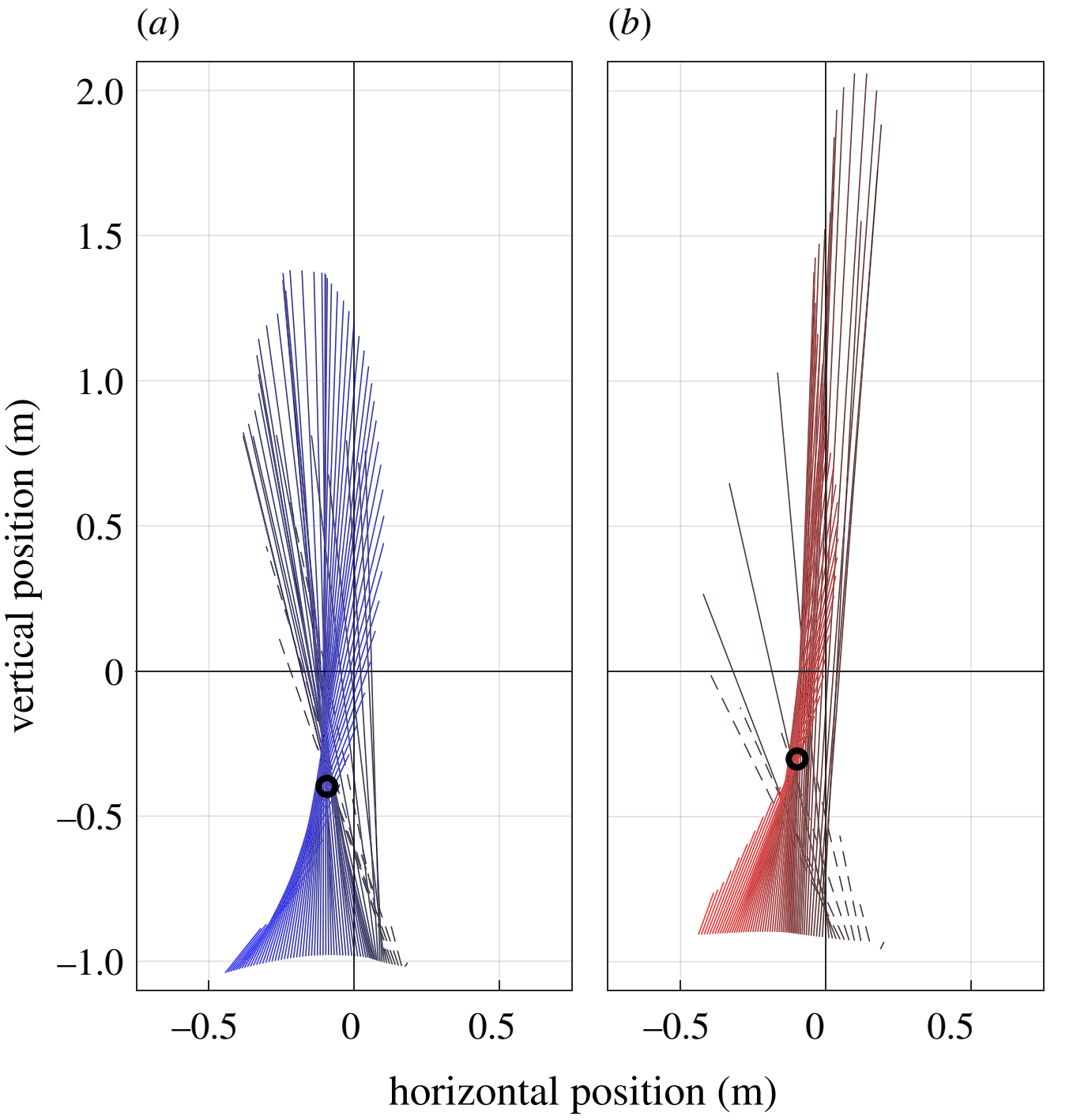


**Figure 3.** Examples of the ground reaction force vectors (GRFs) and the estimated virtual point (VP) for step 0 of V0 (*a*) and C10 (*b*) conditions of the human running experiments. The GRFs and VP are plotted with respect to a CoM-centred, stationary coordinate frame. Lines show the GRFs at different measurement times, originating at the CoP. The 90% dataset consists only of GRF data plotted as solid lines, the 100% dataset includes the entire stance phase GRF data. The black circle indicates the calculated VP for the 90% dataset. (*a*) V0: visible level running, black to blue, (*b*) C10: running with a camouflaged drop of −10 cm, black to red. For each condition, the trial with the spread around the VP nearest to the 50th percentile of all subjects was chosen.

of the stance phase (dashed lines) were neglected in this VP calculation (figure 3). Hence, the VP was computed for 90% and 100% datasets and the results for both VP are given in this section.

The VP in step −1 (pre-perturbed) and step 0 (perturbed) was below the CoM ($p \leq 0.001$, Cohen's D $\leq -1.486$) and between $-38.8 \pm 5.6$ cm and $-24.0 \pm 16.4$ cm (figure 4*a*). For step −1, there were no differences between the ground conditions in the vertical VP position VPz (–31.0 cm) and the $R^2$ (88.7%; table 2). However, the horizontal VP position VPx was 5.5 cm (V10) and 5.7 cm (C10) more posterior in the drop conditions than in the level condition ($p < 0.001$). At step 0, VPx was 4.4 cm more posterior in C10 compared to V0 ($p < 0.028$), and for the 100% dataset 0.8 cm more posterior in V10 than in V0 ($p = 0.038$; table 2). There were only differences in VPz for the 100% dataset, it was 3.6 cm lower in V10 compared to V0 ($p = 0.029$). $R^2$ has the largest value for V0 ($92.0 \pm 2.1$%; 90% dataset) and the smallest one for C10 ($64.1 \pm 8.7$%; 100% dataset, figure 4*b*).

There were no significant differences between the ground conditions in the impulses of step −1 (table 2). For step 0, figure 5 suggests that the vertical GRFs are higher in the step conditions compared to V0, especially for the braking phase. The vertical braking impulse was higher in V10 than in V0 ($p = 0.008$) and in C10 ($p < 0.001$). We observe 2.9 BW peak vertical GRFs in V0, which yield to a vertical braking impulse of 0.69. In V10, the peak vertical GRFs were at 3.4 BW with a braking impulse of 0.83. In C10, the peak was the highest with 3.9 BW, but here, the peak is overlapping with the impact peak and therefore not comparable with that of the visible ground conditions (figure 5). Because of the shorter stance time in C10 (table 2), the braking impulse of 0.63 does not differ from the value of V0 despite the high impact peak. The vertical propulsion impulse of step 0 does not differ significantly between the ground conditions. The amounts of the horizontal braking and propulsion impulses were lower in C10 than in the visible conditions ($p \leq 0.004$). The sum of the horizontal braking and propulsion impulses of step 0 is in all ground conditions around zero. It means that there is no forward acceleration or deceleration.

The vertical CoM position relative to the CoP at the touch-down of step 0 is 3.5 cm higher in the drop conditions compared to V0 ($p < 0.001$) with $104.9 \pm 5.2$ cm and 1 cm higher in C10 than in V10 ($p = 0.019$).

The forward running velocity measured at step 0 does not vary between the experiments V0, V10 and C10, and is within the range of $5.0 \pm 0.5$ m s$^{-1}$. Despite the constant velocity, the stance time and the duty factor of step 0 show a variation between these experiments. The stance time gets shorter ($p$=0.029) and

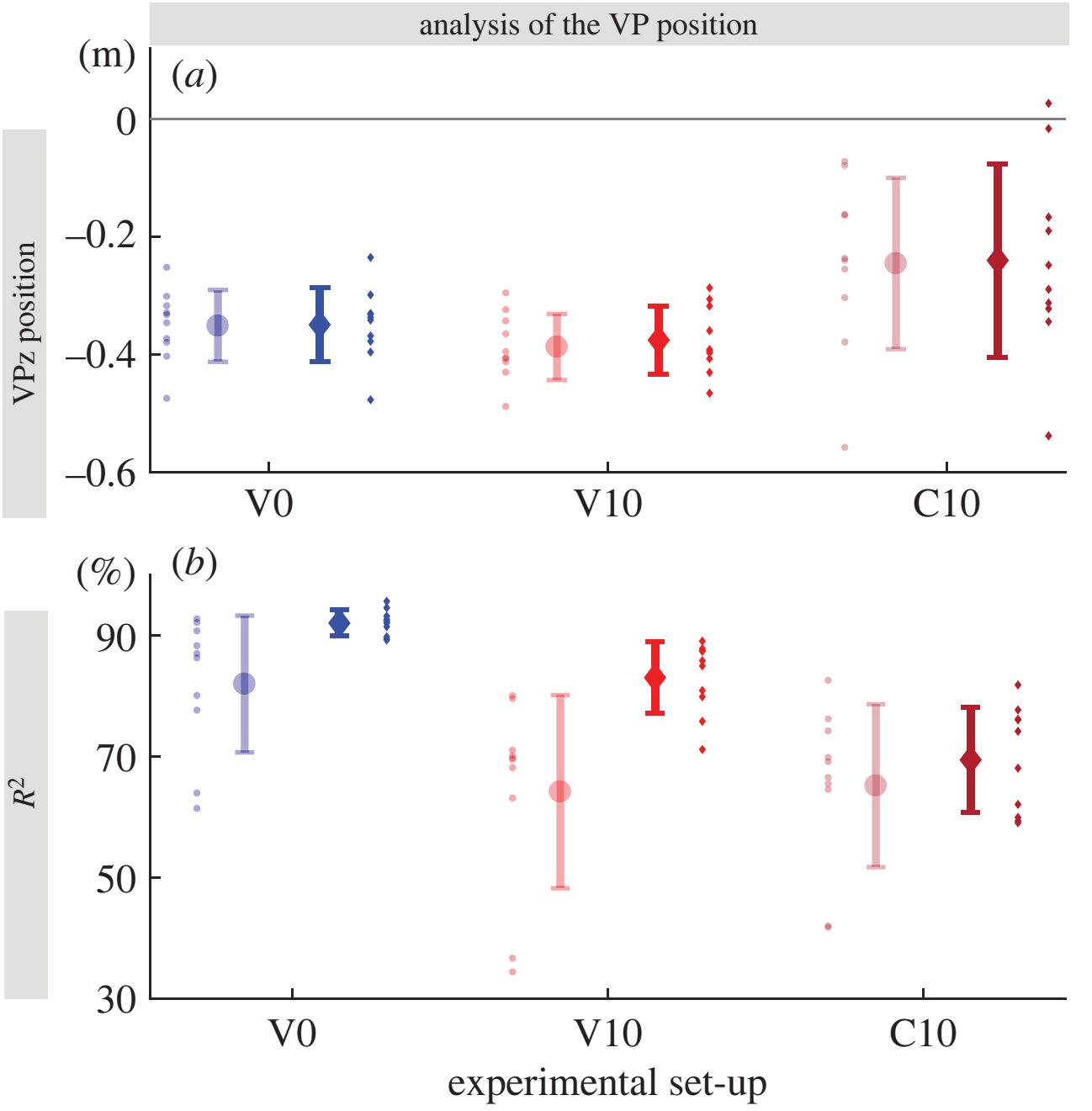


**Figure 4.** Mean ± s.d. of the vertical virtual point position VPz (*a*), and $R^2$ values (*b*) between subjects ($n = 10$) for each ground condition (V0, V10 and C10) for step 0. (*a*) Each small dot is the median over all trials of one condition for one subject. (*b*) $R^2$ represents the ratio of the angle between measured and ideal forces and their variance. Each small dot represents one subject. Transparent circle: 100% dataset, non-transparent diamond: 90% dataset.

the duty factor lower ($p < 0.001$) when running down the visible drop and even shorter and lower when the drop is camouflaged ($p < 0.006$).

## 3.2. Simulation results

In this section, we present our simulation results and our analysis on how the VP reacts to step-down perturbations. The simulation gaits are generated for 5 m s$^{-1}$ running with a VP target −30 cm below the CoM ($VP_{BL}$), which correspond to the estimated values of our experiments in §3.1.

The temporal properties of the base gait for the level running are given in table 3, where the duty factor is calculated as 26.2% with a stance phase duration of 0.16 s. The CoM trajectory of the base gait is shown in figure 7$a_0$ and its respective GRF vectors are plotted with respect to a hip centred stationary coordinate frame in figure 7$b_0$.

The base gait is subjected to step-down perturbations of $\Delta z = [-10,\ -20,\ -30,\ -40\,\text{cm}]$ at step 0. The leg angle controller in equation (2.5) and VP angle controller in equation (2.6) update on a step-to-step basis, therefore are informed about the deviation from the base gait at the beginning of step 1. At step 0, the state of the CoM at leg touch-down diverges from the equilibrium conditions: the trunk pitch angle is smaller (i.e. smaller trunk lean), and vertical speed is higher (see dark grey lines in appendix A.3., figure 13*a*,*c*). The VP position relative to the hip shifts downwards, as seen with circle marker in figure 7$c_1$–$c_4$. The perturbed state leads to an increase in trunk angular excursion during the stance, whereas the step ends with a higher forward speed, smaller trunk lean, and higher trunk angular velocity (see dark grey lines in appendix A.3., figure 13*a*,*b*,*d*). At step 1, the leg angle at touch-down is adjusted to a flatter angle and the VP angle to a larger angle (i.e. VP rotates clockwise). The VP position relative to the hip joint shifts backwards, as seen with dark cross marker in figure 7$c_1$–$c_4$. The backward $VP_{BL}$ shift helps to restore the desired trunk lean and leads to a more pronounced forward trunk motion at step 1 (see red lines in appendix A.3., figure 13*a*). This restoring behaviour can also be inferred from the absence of a counterclockwise rotation towards the leg take-off, i.e. the GRF vectors are not coloured teal towards leg take-off in figure 7$b_1$–$b_4$, in contrast to figure 7$b_0$. We see that the $VP_{BL}$ is able to counteract the step-down perturbations in the following steps by using only local controllers for the VP angle (equation (2.6)) and the leg angle (equation (2.5)), as shown in figure 7$a_1$–$a_4$. As we increase the magnitude of the step-down perturbations, we decrease the coefficients $k_{\dot{x}}$, $k_{\dot{x}_0}$ in the leg angle control, so that the speed correction is slower and the postural control is prioritized (see appendix A.2). The generated gaits are able to converge to the initial equilibrium state (i.e. the initial energy level) within 15 steps after the step-down perturbation at step 0.

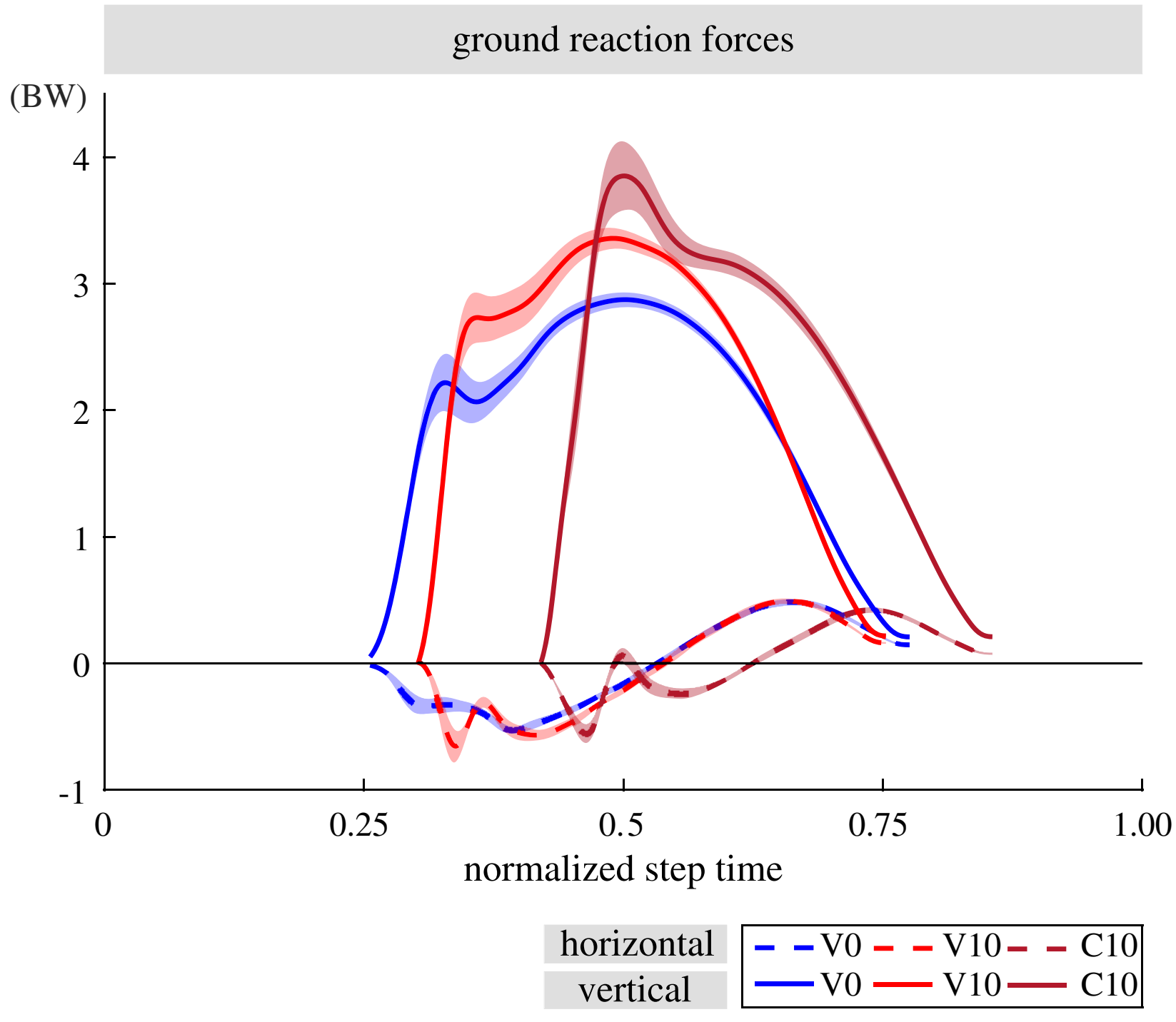


**Figure 5.** The ground reaction forces (GRFs) of step 0 for human running experiments V0 (blue), V10 (red) and C10 (brown). The GRFs are normalized to body weight (BW) of the subjects ($n = 10$). The mean values of the vertical and horizontal GRFs are plotted with solid and dashed lines, respectively. The $\pm$ standard error is shown with the shaded area. For the C10 condition, the vertical GRF peak coincides with the peak caused by the impact peak forces. The duty factor of the V0 condition is 26.7 $\pm$ 2.0%, whereas it is 24.8 $\pm$ 1.6% for the V10 condition and 22.4 $\pm$ 1.5% for the C10 condition.

**Table 3.** Gait properties of the simulated trajectories. (In the presence of step-down perturbations, the $VP_{BL}$ method is able to bring the system back to its initial equilibrium state. Therefore, the gait properties are the same for the even ground and perturbed terrain, after reaching the steady-state condition.)

| property | unit | value | property | unit | value |
|---|---|---|---|---|---|
| duty factor | % | 26.2 | VP angle | ° | −180 |
| stance time | s | 0.16 | trunk angular excursion | ° | 4.45 |
| forward speed | m s$^{-1}$ | 5 | leg angle at touch-down | ° | 66 |

### 3.2.1. Energy regulation

In order to assess the response of the VP controller, we plot the VP position with respect to a hip centred non-rotating coordinate frame that is aligned with the global vertical axis, as it can be seen in figure 7$c_1$–$c_4$. For a $VP_{BL}$ target, a backward shift in VP position indicates an increase in the negative hip work.

The step-down perturbation at step 0 increases the total energy of the system by the amount of potential energy introduced by the perturbation, which depends on the step-down height. The position of the VP with respect to the hip shifts downwards by 0.5–1.9 cm depending on the drop height (see circle markers in figure 7$c_1$–$c_4$). Consequently, the net hip work remains positive and its magnitude increases by 0.7 to 1.7 fold[2] (see solid lines in figures 6*c* and 14*c*). The leg deflection increases by 0.95 to 3 fold, whose value is linearly proportional to the leg spring energy as $E_{SP} = 1/2\,k\,\Delta l_L^2$ (see solid lines in figures 6*a* and appendix A.4., 14*a*). The leg damper dissipates 1.5 to 6 fold more energy compared to its equilibrium condition (see solid lines in figures 6*b* and appendix A.4., 14*b*).

The reactive response of the VP starts at step 1, where the target VP is shifted to backwards by 1.2–2.8 cm and downwards by 0.6–2.9 cm depending on the drop height (see cross markers in figure 7*c*). The backward shift in VP causes a 1.4 to 3.8 fold increase in the negative hip work, and the *net* hip work

[2]For quantities A and B, the fold change is given as (B--A)/A.

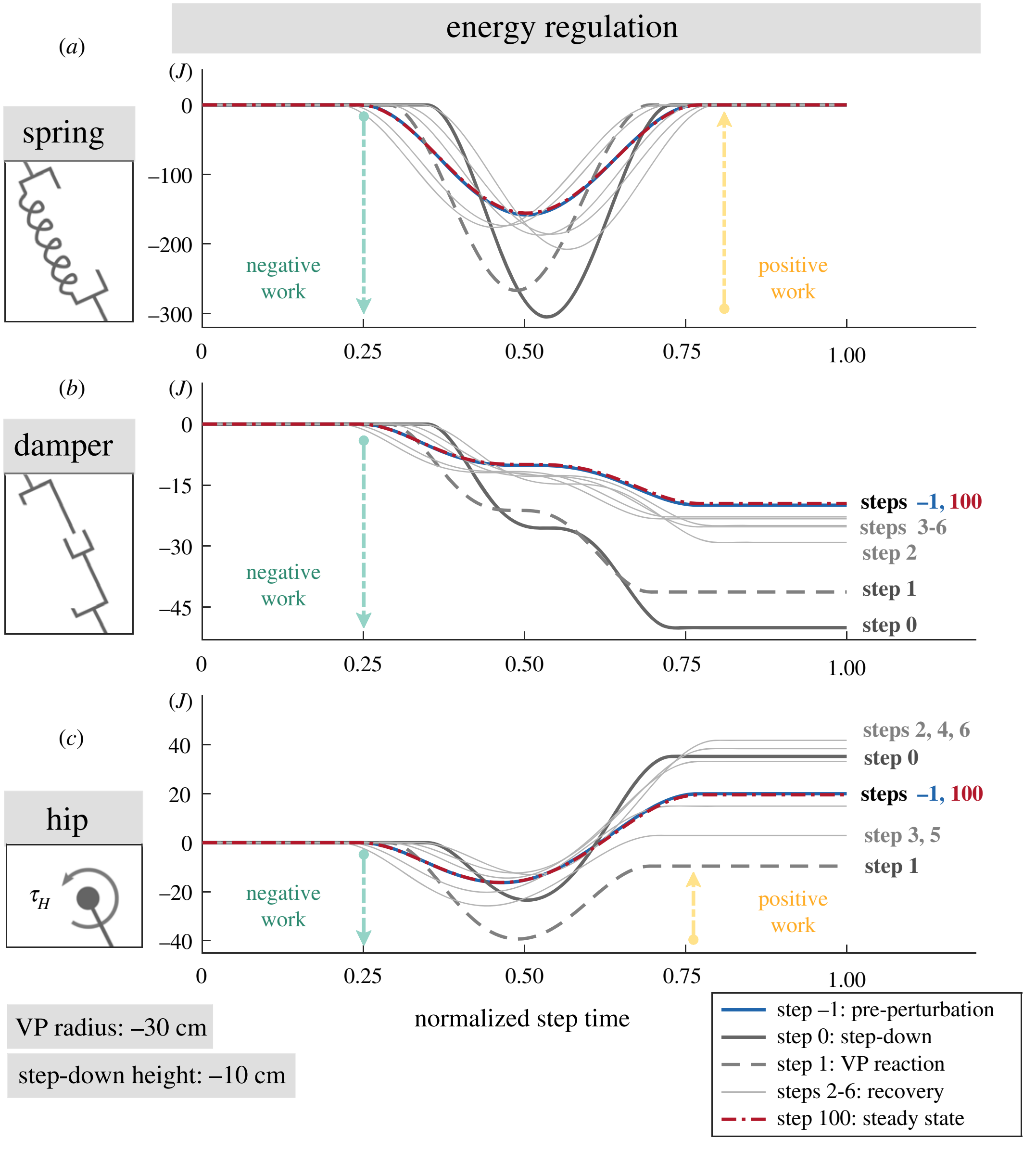


**Figure 6.** The energy levels for the leg spring (*a*), leg damper (*b*) and hip actuator (*c*) for −10 cm step-down perturbation. The step-down perturbation at step 0 increases the energy of the system, which causes an increase in leg deflection and a larger fluctuation in spring energy (*a*, solid dark grey line). The leg damper dissipates more energy and the hip actuator injects more energy than during its equilibrium condition (*b–c*, solid dark grey line). Starting with step 1, the VP begins to react to the energy change and the hip actuator starts to remove energy from the system (*c*, dashed line). In the following steps (solid grey line), the hip regulates the energy until the system reaches to the initial equilibrium state (solid blue line). Extended plots for the step-down height of $\Delta z = [-20, \ -30, \ -40\,\text{cm}]$ can be found in appendix A.4.

becomes negative (see dashed lines in figures 6*c* and appendix A.4., 14*c*). In other words, the hip actuator starts to remove energy from the system. As a result, the trunk leans more forward during the stance phase (see yellow coloured GRF vectors in figure 7*b*). The leg deflects 0.7 to 2.3 fold more than its equilibrium value, and the leg damper removes between 1 and 4.1 fold more energy. However, the increase in leg deflection and damper energy in step 1 are lower in magnitude compared to the increase in step 0. In step 1, we see the $VP_{BL}$'s capability to remove the energy introduced by the step-down perturbation.

In the steps following step 1, the target VP position is continued to be adjusted with respect to the changes in the trunk angle at apices, as expressed in equation (2.6) and shown with cross markers in figure 7*c*. The VP position gradually returns to its initial value, and the gait ultimately converges to its initial equilibrium, see coinciding markers diamond, rectangle in figure 7*c*. During this transition,

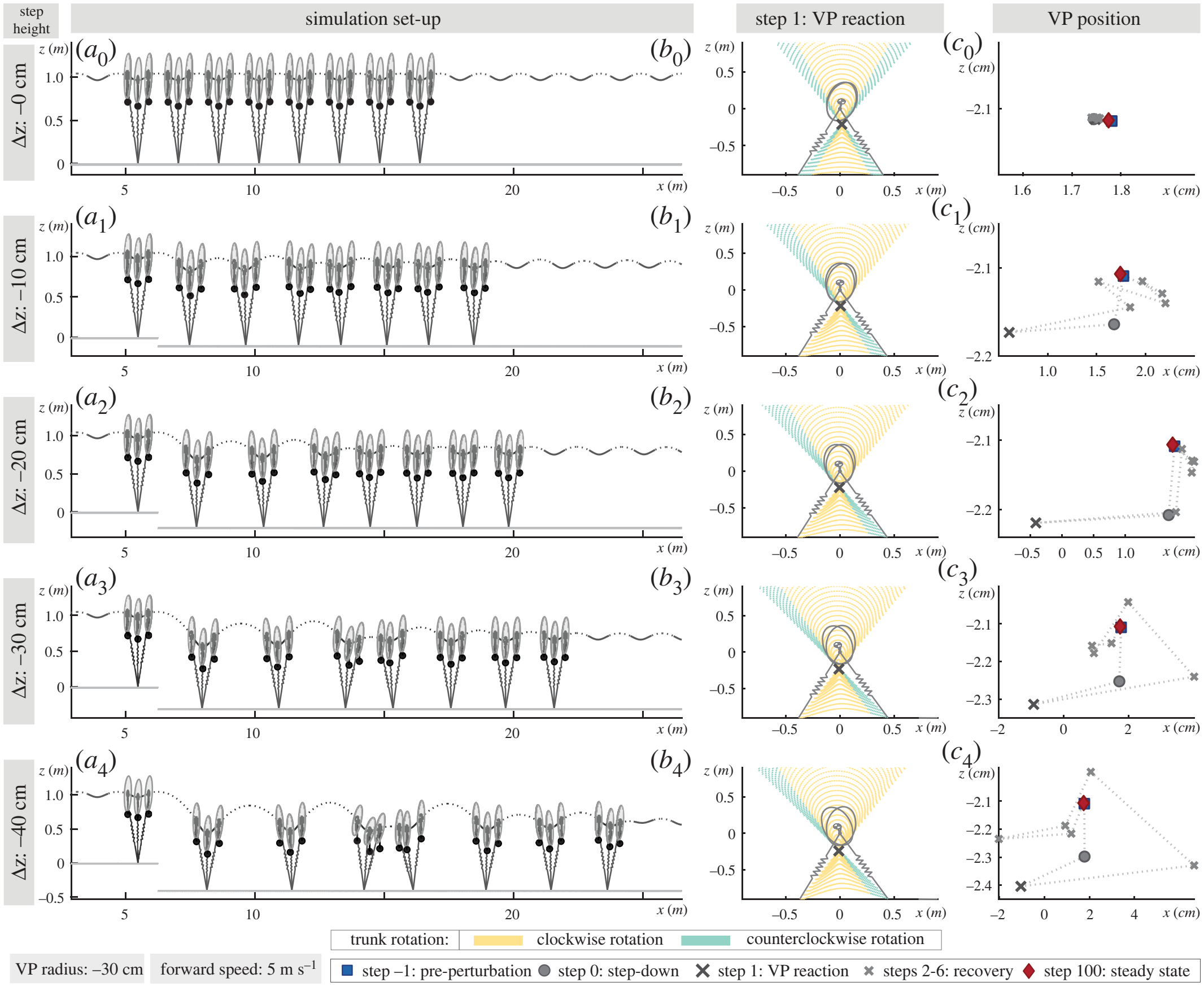


**Figure 7.** The analysis begins with a base gait $a_0$ based on the human running experiment V0, which has a VP target of −30 cm with a forward speed of 5 m s$^{-1}$. This base gait is then subjected to step-down perturbations of $\Delta z = [-10, \ -20, \ -30, \ -40\,\mathrm{cm}]$ at step 0. The −10 cm perturbation corresponds to V10–C10 of the human running experiments. The model state at touch-down, mid-stance and take-off instances of steps −1 to 6 are drawn in $a_0$–$a_4$ to display the changes in the trunk angle. At the perturbation step, the VP position shifts downwards with respect to a hip centred stationary coordinate frame (circle in $c_1$–$c_4$). VP$_{BL}$ counteracts to the perturbation at step 1 with a backward shift, which depletes the energy added by the stepping down (dark cross marker in $c_1$–$c_4$). The GRF vectors of step 1 causes a forward trunk lean of 5 to 10°, which is shown in $b_1$–$b_4$. In the following steps, VP position is regulated to achieve the energy balance (cross marker), and gaits ultimately reach to the equilibrium state. The equilibrium state is given in table 3. A single gait involves 100 successful steps (diamond markers in figure $c_1$–$c_4$).

the energy interplay between the hip and leg successfully removes the energy added to the system, as shown in figure 6*b*,*c* and in appendix A.4., figure 14*b*,*c* for larger step-down perturbation magnitudes.

### 3.2.2. Ground reaction force analysis

The energy increment owing to the step-down perturbation and the energy regulation of the VP$_{BL}$ control scheme can also be seen in the GRF and impulse profiles.

The peak vertical GRF magnitude of the equilibrium state is 3 BW. It increases to 4.2–6.1 BW at step 0 with the step-down (figure 8*c*,*a*). The peak magnitude decreases gradually to its initial value in the following steps, indicating that the VP is able to bring the system back to its equilibrium. In a similar manner, the normalized vertical impulse increases from 1 to 1.4–2.2 at step 0 (see circle marker in figure 8*d*,*b*) and decreases to 1 in approximately 15 steps.

The peak horizontal GRF magnitude of the equilibrium state amounts to 0.6 BW. It increases to 0.9–1.4 BW at step 0 (figures 8*a* and appendix A.5., 15*a*). The sine shape of the horizontal GRF and its peak magnitude depend on the change in VP position. Therefore, the horizontal GRF impulse provides more information. The net horizontal GRF impulse is zero at the equilibrium state (see rectangle in figures 8*b* and appendix A.5., 15*b*). It becomes positive at the step-down perturbation (circle), leading to a net horizontal acceleration of the CoM. Consequently, the forward speed increases at the end of step 0 (see dark grey lines in appendix A.3., figure 13*b*). In step 1, the VP$_{BL}$ and leg touch-down angle are adjusted with respect to the change in the state, which leads to a negative net horizontal GRF impulse (dark cross marker) and decelerates the

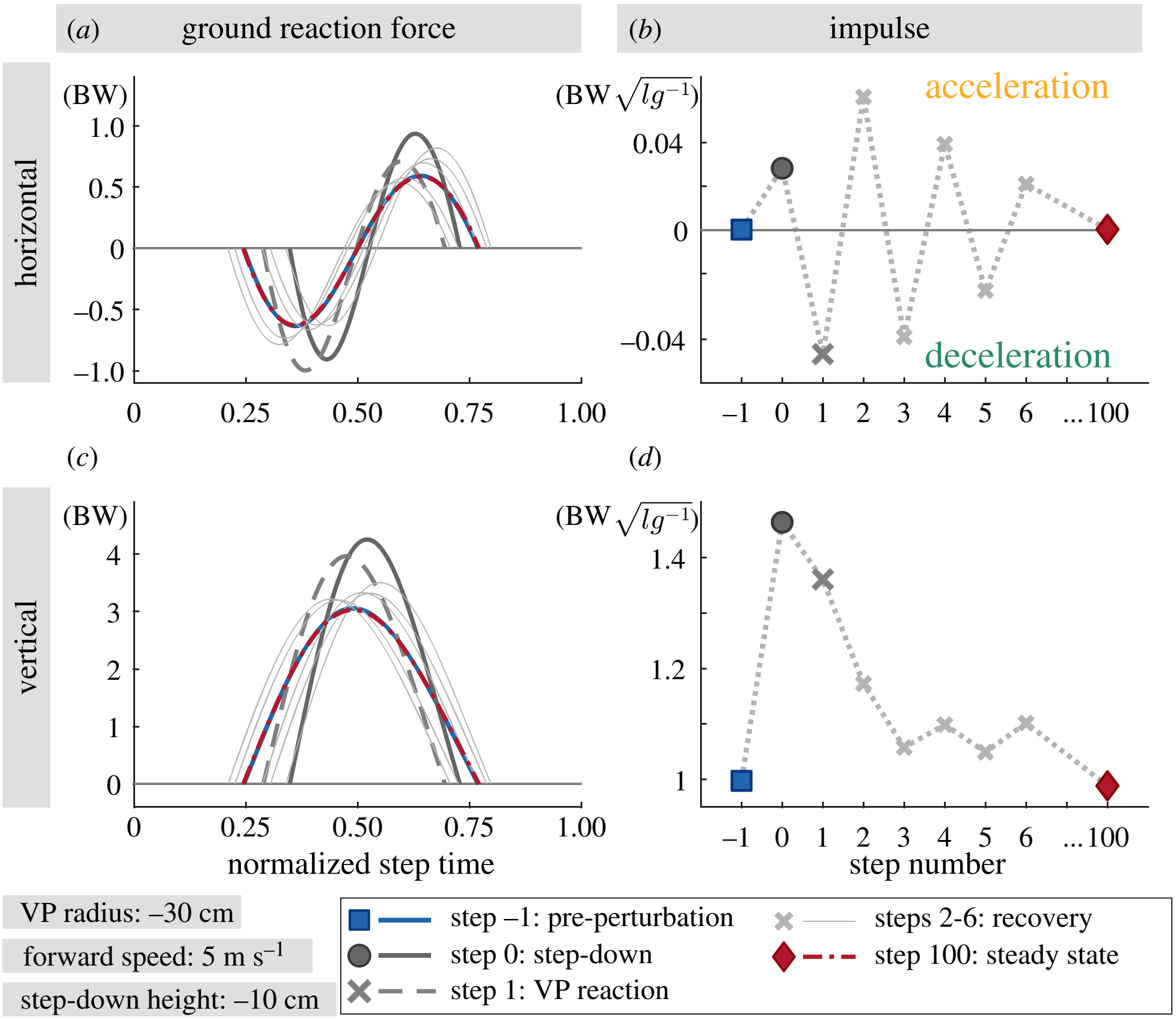


**Figure 8.** Numerical simulation results: the ground reaction forces (*a*,*c*) and the corresponding net impulses (*b*,*d*) for −10 cm step-down perturbation. The GRFs are normalized to body weights (BW), whereas the impulses are normalized to their BW$\sqrt{l/g}$ values. The effect of the $VP_{BL}$ control can be seen in the horizontal GRF and impulse. $VP_{BL}$ alters the net horizontal impulse, and causes either net horizontal acceleration or deceleration after the step-down perturbation. Consequently, the excess energy introduced by the perturbation is removed from the system. The vertical GRF and impulse increase with the perturbation and decrease gradually to its equilibrium value approximately within 15 steps. Extended plots for the step-down height of $\Delta z = [-20, \ -30, \ -40\,\text{cm}]$ can be found in appendix A.5.

body (see red lines in appendix A.3., figure 13*b*). In the following transient steps, the leg and VP angle adjustment yields successive net accelerations and decelerations (cross marker) until the system returns to its equilibrium state (diamond), where the net horizontal GRF impulse and forward acceleration is zero.

## 4. Discussion

In this study, we performed an analysis of experimental and simulation results regarding the force direction patterns during human-level running, and running onto a visible or camouflaged step-down. Our experimental results show that humans tend to generate a VP below the CoM ($VP_B$) for all terrain conditions. Our simulations support these experimental observations, and show that the $VP_B$ as a controller can cope with step-down perturbations up to 0.4 times the leg length. In this section, we will address the VP location in connection with the gait type, and will discuss how our experimental results compare to our simulation results for the running gait.

### 4.1. Virtual point quality and location in human gait

In the first part, we discuss the validity of a VP estimated from the GRF measurements of the human running. We only consider step 0 of the 90% dataset, because the 100% dataset is biased by the additional effects of the impact forces and has low $R^2$ values [7]. In the second part, we discuss how the VP position is correlated to the gait type.

To determine the quality of the VP estimation, we used the coefficient of determination $R^2$. In our experiments, the $R^2$ values for level running are high, where $R^2 \approx 92\%$ (see V0 in figure 4*b*). The

values of the $R^2$ get significantly lower for the visible drop condition, where $R^2 \approx 83\%$ (see V10 in figure 4*b*). On the other hand, the $R^2$ of the camouflaged drop conditions are even lower than for the visible drop conditions, where $R^2 \approx 69\%$ (see C10 in figure 4*b*). An $R^2$ value of $\approx$70% is regarded as 'reasonably well' in the literature ([37], p. 475). Based on the high $R^2$ values, we conclude that the measured GRFs intersect near a *point* for the visible and camouflaged terrain conditions. We can also confirm that this point is as hypothesized below the CoM ($VP_B$), as the mean value of the estimated points is −32.2 cm and is significantly below the CoM.

We find a difference in the estimated VP position between the human walking and our recorded data of human running. The literature reports a VP above the CoM ($VP_A$) for human walking gait [1–3,5], some of which report a $VP_A$ in human running as well [7,10]. By contrast, our experiments show a $VP_B$ for human-level running at 5 m s$^{-1}$ and running over a visible or camouflaged step-down perturbation. Previous studies only reported single trials of single subjects and no statistical analysis. We also did observe a few trials as outliers with a VP above the CoM, which are statistically not significant. Additionally, different ways for cropping the contact phase were considered in the previous studies, which affects the estimation of the vertical VP position. Here, we consistently remove the first 10% of the contact phase. Our experimental set-up and methodology are identical to [5], which reports results from human walking. Thus, we can directly compare the $R^2$ values for both walking and running. The $R^2$ value of the level running is 6 percentage points lower than the $R^2$ reported in [5] for level walking. The $R^2$ value for V10 running is 15 percentage points lower than V10 walking, whereas the $R^2$ for C10 running is up to 25 percentage points lower compared to C10 walking. In summary, we report that the spread of the $R^2$ is generally higher in human running at 5 m s$^{-1}$, compared to human walking.

## 4.2. Experiments versus model

In this section, we discuss how well the TSLIP simulation model predicts the CoM dynamics, trunk angle trajectories, GRFs and energetics of human running. A direct comparison between the human experiments and simulations is possible for the level running. The V0 condition of the human experiments corresponds to step −1 of the simulations (also to the base gait). Overall, we observe a good match between experiments and simulations for the level running (see figures 9–11). On the other hand, a direct comparison for the gaits with perturbed step is not feasible owing to the reasons given in §4.3 in detail. Here, we present perturbed gait data to show the extent of the similarities and differences between the V10 and C10 conditions of the experiments and step 0 and 1 of the simulations.

Concerning the CoM dynamics, the predicted CoM height correlates closely with the actual CoM height in level running, both of which fluctuate between 1.05–1 m with 5 cm vertical displacement (figure 9$a_1$–$a_2$). The vertical displacement of the CoM is larger for the perturbed step, where the CoM height alternates between 1–0.9 m in the experiments (figure 9$a_3$) and 1.05–0.85 m in the simulations (figure 9$a_4$). The differences can be attributed to the visibility of the drop. Human runners visually perceiving changes in ground level and lowered their CoM by about 25% of the possible drop height for the camouflaged contact [41]. The mean forward velocity at leg touch-down is 5.2 m s$^{-1}$ in the experiments (figure 9$b_1$). In the simulations, the leg angle controller adjusts the forward speed at the apex to a desired value. We set the desired speed to 5 m s$^{-1}$ (figure 9$b_2$), which is the mean forward velocity of the step estimated from the experiments. For level running, both the experiments and simulations show a 0.2 m s$^{-1}$ decrease in forward velocity between the leg touch-down and mid-stance phases (figure 9$b_1$–$b_2$). As for the perturbed running, human experimental running shows a drop in forward speed of 4.5% for V10, and 0.1% for the C10 condition (figure 9$b_3$). Namely, there is no significant change in forward velocity during the stance phase for the C10 condition. The simulation shows a drop in forward speed of 9.5% for step 0, and 11.1% in step 1 (figure 9$b_4$).

The trunk angle is the least well-predicted state, since the S-shape of the simulated trunk angle is not recognizable in the human running data (figure 9$c_1$–$c_2$). One of the reasons may be the simplification of the model. The flight phase of a TSLIP model is simplified as a ballistic motion, which leads to a constant angular velocity of the trunk. The human body, on the other hand, is composed of multiple segments, and intra-segment interactions lead to more complex trunk motion during flight phase. Furthermore, the model does not distinguish between the trunk and whole-body dynamics [42]. The large variance observed in the trunk angle trajectories between different subjects and trials might obscure small trunk angle tendencies, particularly for the C10 condition. Consequently, the mean trunk angle profiles do not provide much information about the trunk motion pattern, especially for the perturbed step for C10. Therefore, we cannot clarify to what extent the VP position is used for regulating the trunk motion in humans. However, a trend of trunk moving forward is visible in both simulation and

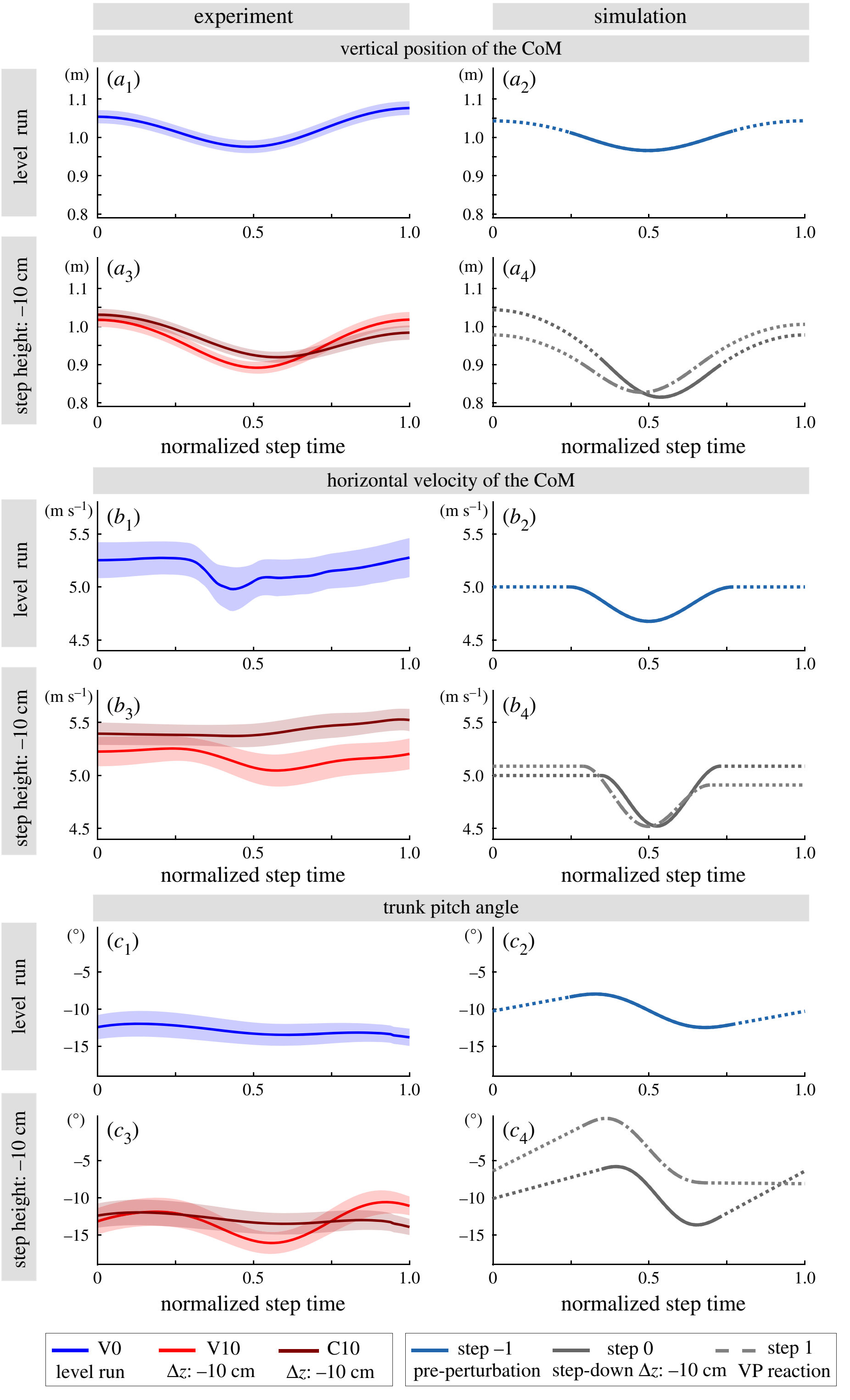


**Figure 9.** The CoM height (*a*), horizontal CoM velocity (*b*), and trunk angle (*c*) for step 0 of the experiments V0, V10 and C10 are shown on the left, and the steps −1, 0 and 1 of the simulation are shown on the right column. The mean is shown with a line and the standard error is indicated with the shaded region. The standard error equals to the standard deviation divided by the square root of number of subjects. The TSLIP model is able to predict the CoM height and forward speed. Its prediction capability is reduced for the trunk motion, as the flight phase involves ballistic motion and the trunk angular velocity is constrained to be constant.

experiments. The mean trunk angular excursion at step 0 of the experiments is 1.8° for V0, 5.5° for V10 and 1.9° for the C10 condition (figure 9$c_1$–$c_3$). The S-shaped pattern of the trunk motion becomes more perceivable in the experiments with a visible perturbed step (figure 9$c_3$). In the simulations, the trunk angular excursion is set to 4.5° for level running based on [27,39,40]. The magnitude of the trunk

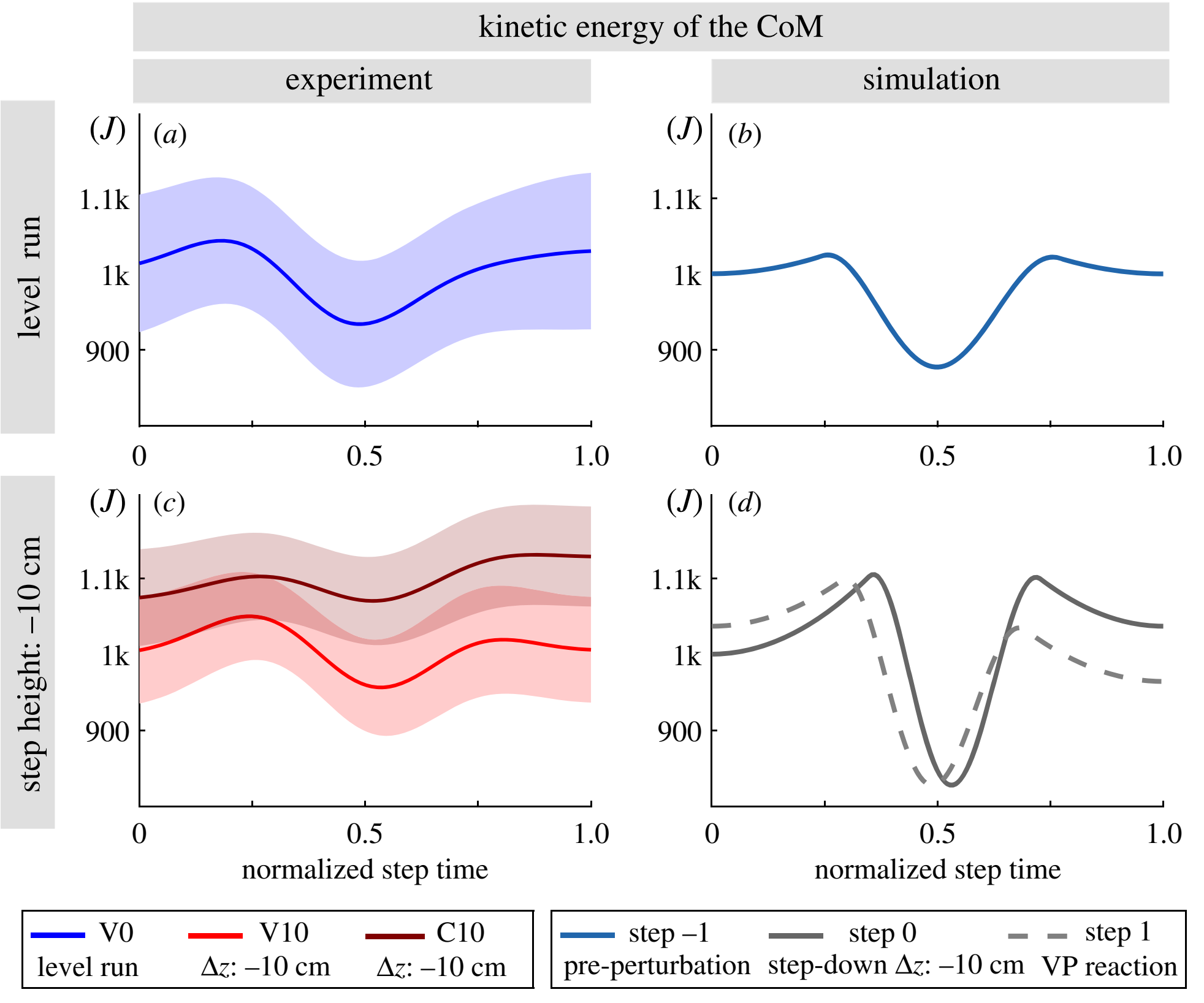


**Figure 10.** Kinetic energy of the CoM for the human running experiments (left) and simulated model (right). The mean is shown with a line and the standard error is indicated with the shaded region. The standard error equals to the standard deviation divided by the square root of number of subjects. The TSLIP model is able to predict the kinetic energies for the unperturbed and visible perturbed step well. The simulation yields larger energy fluctuations during the stance phase compared to experiments. Experiments with camouflaged perturbation (C10) yield higher mean kinetic energy compared to the ones with visible perturbations (V10).

rotation at the perturbation step is higher in simulations, and amounts to 7.8° at step 0 and 8.6° at step 1 (figure 9$c_2$–$c_4$).

There is a good agreement between the simulation-predicted and the recorded GRFs for level running. The peak horizontal and vertical GRFs amount to 0.5 BW and 3 BW, respectively, in both experiments and simulations (see figures 5, 8*a*,*d* and appendix A.7., 19). As for the step-down perturbation, the simulation model is able to predict the peak vertical GRF, but the prediction becomes less accurate for the peak horizontal GRF. The peak vertical GRF of the −10 cm step-down perturbation case is 3.5 BW for the V10 condition and 4 BW for the C10 condition, whereas it is 4 BW for the simulation. In the C10 condition, the vertical GRF peak occurs at the foot impact and its peak is shifted in time, to the left. The numerical simulation leads to over-simplified horizontal GRF profiles, in the step-down condition. The human experiments show an impact peak. The experiments have a peak horizontal GRF magnitude of 0.5 BW, which remains the same for all perturbation conditions. By contrast, the peak horizontal GRF increases up to 1 BW in simulations.

In level running the GRF impulses of the experiments and the simulation are a good match (see table 2 and appendix A.5., figures 15*b* and 16*b*). The normalized horizontal impulses for both braking and propulsion intervals are the same at 0.1, while the normalized net vertical impulse in experiments are 15% higher than in the simulation. For the step-down conditions, the simulation predicts higher normalized net vertical impulse values of 1.46 at step 0 and 1.36 at step 1, as opposed to 1.31 for the V10 condition and 1.18 for C10 condition in experiments. The change in the horizontal impulses during the step-down differs significantly between the simulation and experiments. The V10 condition shows no significant change in the horizontal impulses, while in the C10 condition they decrease to 0.04 for breaking and 0.06 for propulsion. By contrast, the simulations show an increase in the horizontal impulses (appendix A.5., figure 15*b*). In particular, for a step-down perturbation of −10 cm, the normalized braking impulse increases to 0.15 at step 0 and 0.18 at step 1, whereas for propulsion it increases to 0.15 and 0.12.

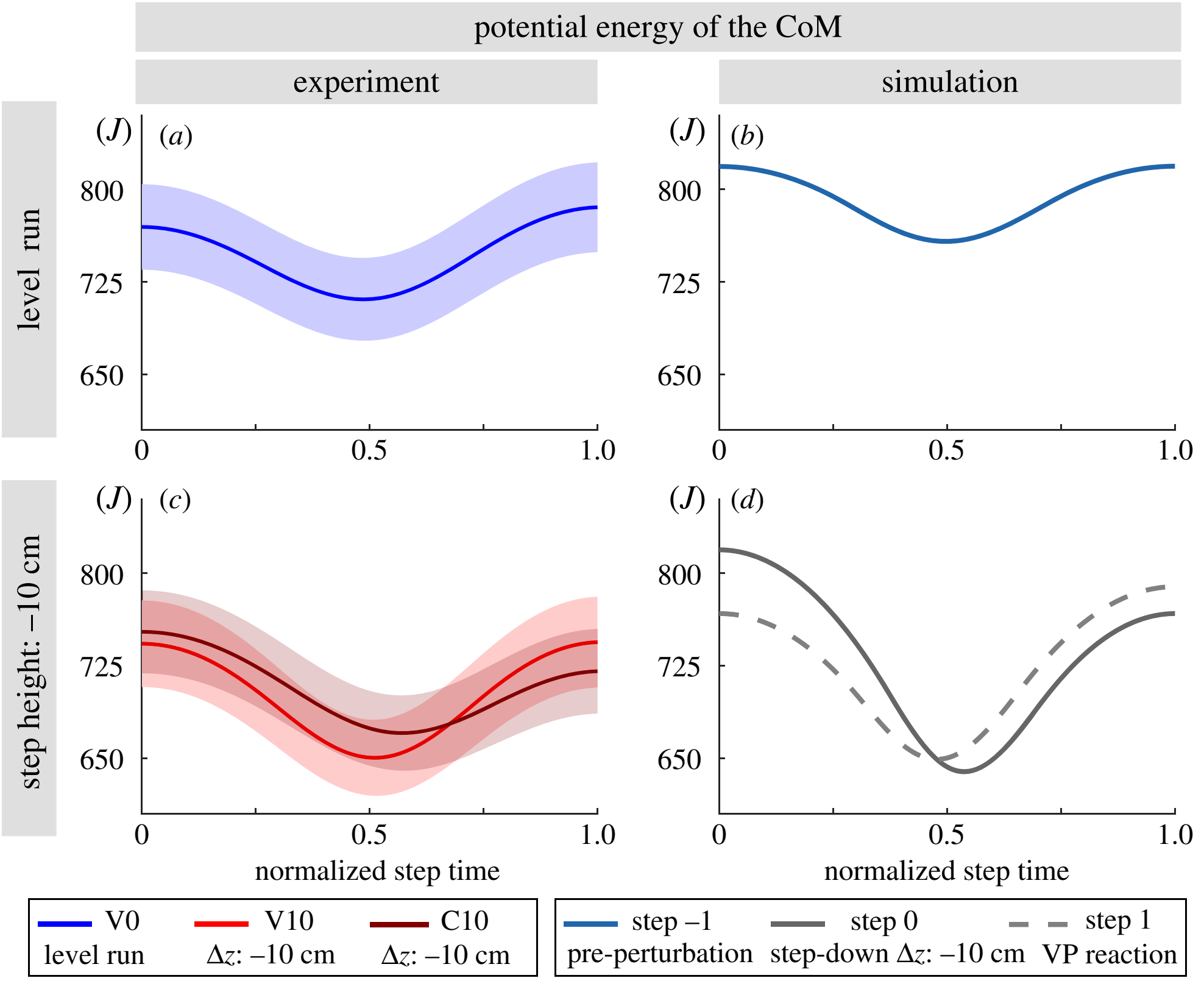


**Figure 11.** Potential energy of the CoM for the human running experiments (left) and simulated model (right). The mean is shown with a line and the standard error is indicated with the shaded region. The standard error equals to the standard deviation divided by the square root of number of subjects. Overall, the TSLIP model predicts the CoM height and its related potential energy well.

The different behaviour we observe in horizontal impulses at step-down for the experiments and simulations may be owing to different leg angles at touch-down. We expect that a steeper leg angle of attack at touch-down would decrease the horizontal and increase the vertical braking impulse. However, we observe with 66° a 9° steeper angle of attack in the simulations for level running than it was reported for V0 for the same experiments [30]. Nevertheless, no corresponding changes in the braking impulses could be observed. On the other hand, in the perturbed condition, the angle of attack is with 66° nearly the same in the simulation and C10, but here the braking impulses differ. However, differences in the definition of the leg also affect the angle of attack. In the literature, an angle of attack between 57° and 82° was reported for level running [16], which fits the model's leg angle of attack. Additionally, no corresponding changes in the braking impulses could be observed. In the perturbed condition, the angle of attack in the experiments is 9° steeper compared to V0, which could be caused by the swing leg retraction, while the angle of attack in the simulation is the same in V0 and C10. Here, also the braking impulses between experiment and simulation differ. Therefore, we conclude that the simulation could potentially be improved by implementing a swing leg retraction as observed in humans [30,43,44].

Another reason for the limited prediction capabilities of the model for step-down drops might be related to the heel strike and impact dynamics during the leg touch-down. The impact peak (i.e. first peak) of the horizontal GRF in the breaking interval is smaller than the active peak (i.e. second peak) for V0, and larger for V10 and C10 conditions (appendix A.7., figure 19$b_0$–$b_1$). In particular, the C10 condition displays a M-shaped horizontal GRF during braking interval, where the impact dynamics becomes dominant. The absence of impact dynamics and associated GRF peaks in the simulation model might contribute to the discrepancy observed in peak GRF magnitudes while stepping down. Therefore, we conclude that additional factors have to be involved in the explanation of the different peaks and impulses between simulation and experiments, and further investigations are needed.

In terms of the CoM energies, there is a good match between the kinetic energies of the experiments and simulations for the unperturbed step (V0 and step −1 in figure 10*a*,*b*). The simulated energies of the perturbed step are closer to the experiments with visible perturbations (V10 and steps 0 and 1 in figure 10*c*,*d*). Human experiments show a drop in kinetic energy of 9% for V10, 3% for C10. The simulation shows a drop in kinetic energy of about 25% for step 0 and step 1. The C10 condition shows a higher

mean kinetic energy compared to visible perturbations and there is no obvious decrease of energy in the stance phase (figure 10*c*).

The potential energy estimate of the simulations lies in the upper boundary of the experiments for the unperturbed step (V0 and step −1 in figure 10*a*,*b*). The experiments with visible and camouflaged perturbations, as well as the TSLIP model, result in similar potential energy curves (figure 10*c*,*d*).

## 4.3. Limitations of this study

The human experiments and the numerical simulations differ in several points, and conclusions from a direct comparison must be evaluated carefully. We discuss details for our choice of human experimental and numerical simulation conditions in this section.

First of all, there is a difference in terrain structure. After passing step 0, the human subjects face a different terrain structure type, compared to the TSLIP simulation model. The experimental set-up is constructed as a pothole: a step-down followed by a step-up. However, an identical step-up in the numerical simulation would require an additional set of controllers to adjust the TSLIP model's leg angle and push off energy. Hence for the sake of simplicity, the TSLIP model continues running on the lower level and without a step-up. After the step-down perturbation, the simulated TSLIP requires several steps to recover. An experimental set-up for an equivalent human experiment would require a large number of force plates, which were not available here.

In the V10 condition, the subjects have a visual feedback and hence the prior knowledge of the upcoming perturbation. This additional information might affect the chosen control strategy. In particular, because there is a step-up in the human experiments, subjects might account for this upcoming challenge prior to the actual perturbation.

In the C10 condition, some subjects might prioritize safety in the case of a sudden and expected drop, and employ additional reactive strategies [45]. By contrast, the simulations with a VP controller cannot react to changes during the step-down and only consider the changes of the previous step when planning for the next.

Furthermore, in the human experiments, we cannot set a step-down higher than −10 cm owing to safety reasons, especially in the camouflaged setting. Instead, we can evaluate these situations in numerical simulations and test whether a hypothesized control mechanism can cope with higher perturbations. However, one has to keep in mind that the TSLIP model that we use in our analysis is simplified. Its single-body assumption considers neither intra-segment interactions, nor leg dynamics from impacts and leg swing. A future model can be improved by including swing leg dynamics, collusion dynamics and ankle torque to capture the heel-strike and ankle push-off effects [46–48]. Finally, our locomotion controller applied does not mimic specific human neural locomotion control or sensory feedback strategy.

## 4.4. Potential uses for the virtual point

Our previous study in [23] offers an explanation why different VP behaviours can be observed in human-level running, by suggesting that different VP targets lead to a trade-off between the energy requirements of the leg and hip. In particular, a VP below the CoM ($VP_{BL}$) with prominent forward trunk motion at ground contact might indicate weaker leg actuation (e.g. caused by injuries), whereas a $VP_B$ closer to the CoM or a $VP_A$ might indicate weaker hip actuation (e.g. caused by hip extensor strength deficit). Robots and rehabilitation devices could be designed with smaller actuators, after adopting VP positions leading to lower joint loads.

If the VP is an existing function in human gait, the VP-based controllers establish biomechanically similar patterns to humans. Consequently, VP-based controllers can provide natural reference trajectories for exoskeletons to assist the human gait with a greater efficiency. For example, VP inspired controllers implemented in the lower limb exoskeleton LOPEZ II [49] and soft passive exosuit in [50] are able to reduce leg muscle activations and decrease the metabolic cost by 10% and 4%, respectively. Our current work can provide the foundation for a VP-based control approach to assist the human gait in the presence of step-down perturbations.

# 5. Conclusion

In this work, we investigated the existence and position of a VP in human running gait, and analysed the implications of the observed VP location to postural stability and energetics with the help of a numerical simulation.

In addition to level running, we also inquired into the change of VP position when stepping down on a −10 cm visible or camouflaged drop. Our novel results are twofold: first, the ground reaction forces focus around a point that is −30 cm below the CoM for the human running at 5 m $s^{-1}$. The VP position does not change significantly when stepping down a visible or camouflaged drop of −10 cm. Second, the TSLIP model simulations show that a VP target below the CoM is able to stabilize the body against step-down perturbations without any need to alter the state or model parameters.

Ethics. The experiment was approved by the local ethics committee and was in accordance to the Declaration of Helsinki.

Data accessibility. Kinetic and kinematic data of the human running experiments are available from the figshare repository: https://doi.org/10.6084/m9.figshare.12034350.v1.

Authors' contribution. J.V. performed the gait analysis for human running. Ö.D. shares first authorship owing to generation of the simulation model and analysis. Both J.V. and Ö.D. wrote the manuscript and all authors discussed the results and contributed to the final manuscript.

Competing interests. We declare we have no competing interests.

Funding. The human running project was supported by the German Research Foundation (Bl 236/21 to Reinhard Blickhan and MU 2970/4-1 to R.M.). The International Max Planck Research School for Intelligent Systems (IMPRS-IS) supported Ö.D. This work was partially made possible thanks to a Max Planck Group Leader grant awarded to A.B.-S. by the Max Planck Society.

Acknowledgements. We thank Martin Götze and Michael Ernst for supporting the experiments.

# Appendix A

## A.1. Simulation: spring loaded inverted pendulum model extended with a trunk parameters

The TSLIP model parameters are presented in table 4 (see [23] for the parameters for the human model and [25] for the avian model).

**Table 4.** TSLIP model parameters.

| name | symbol | units | literature | chosen | reference |
|---|---|---|---|---|---|
| mass | $m$ | kg | 60–80 | 80 | [21] |
| moment of inertia | $J$ | kg $m^2$ | 5 | 5 | [21,42] |
| leg stiffness | $k$ | kN $m^{-2}$ | 16–26 | 18 | [15,21] |
| leg length | $l$ | m | 1 | 1 | [21] |
| leg angle at TD | $\theta_L^{TD}$ | ° | 78–71 | $f_H(\dot{x})$ | [15,21] |
| distance Hip-CoM | $r_{HC}$ | m | 0.1 | 0.1 | [21,51] |

## A.2. Simulation: flowchart for leg angle and virtual point angle control





The linear controller for the leg angle $\theta_L$ and VP angle $\theta_{VP}$ is presented in figure 12. The leg angle control coefficients ($k_{\dot{x}}\, k_{\dot{x}_0}$) in equation (2.5) are decreased from (0.25, $0.5k_{\dot{x}}$) to (0.2, $0.3k_{\dot{x}}$), as the step-down height is increased from −10 cm to −40 cm. The reduction of the coefficients slows down the adjustment of the forward speed, and enables us to prioritize the postural correction in the presence of larger perturbations.

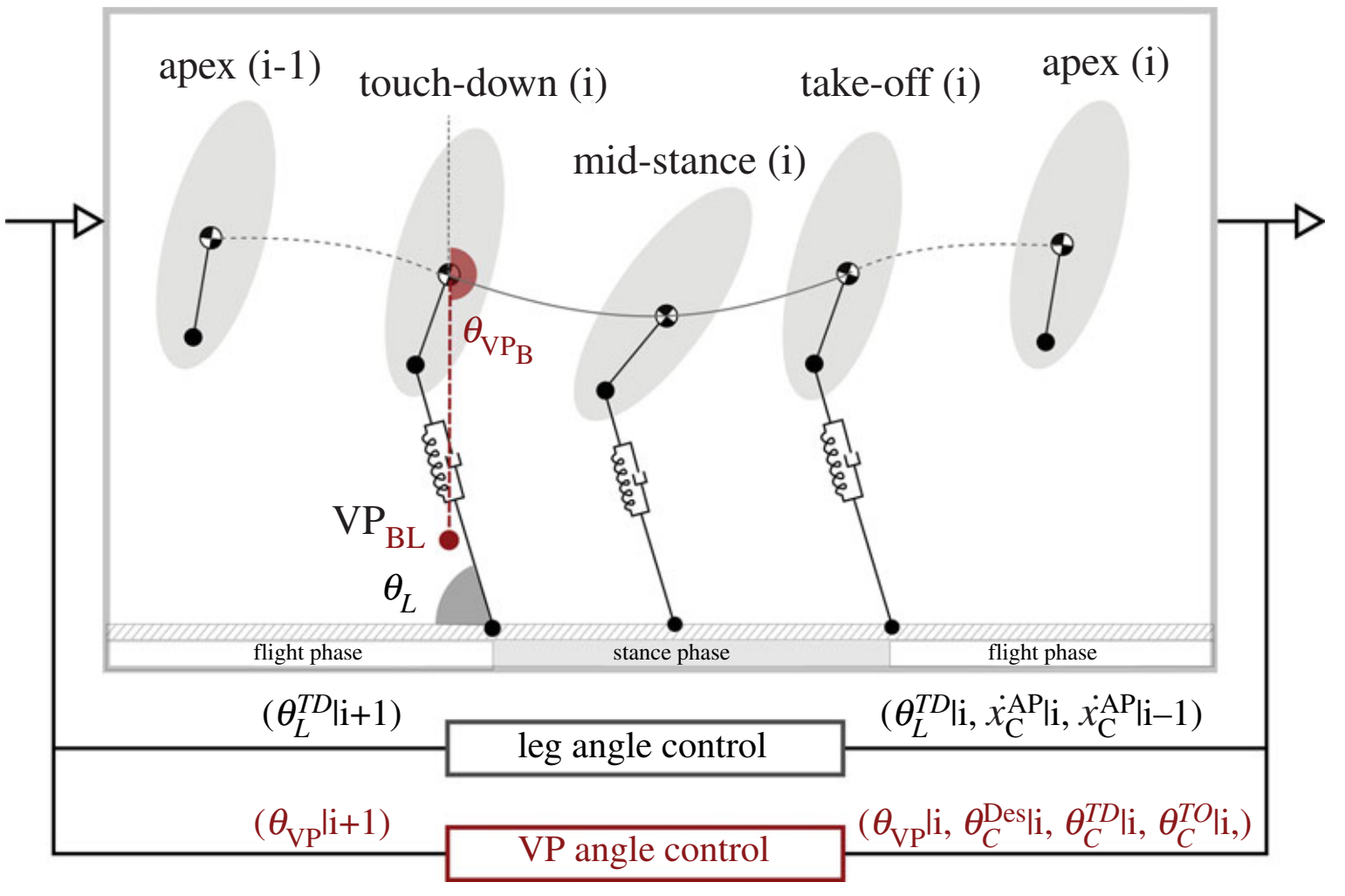


**Figure 12.** The linear feedback control scheme for the leg angle in equation (2.5) and the VP angle in equation (2.6) are presented. Both controllers update step-to-step at the apex event where the CoM reaches to its maximum height.

## A.3. Simulation: the centre of mass state

The centre of mass trajectory of the simulated gaits are plotted in figure 7, where we can infer its position and height. In figure 13, we provide the time progression of the remaining state parameters: the trunk pitch angle, forward speed, vertical speed and trunk angular velocity.

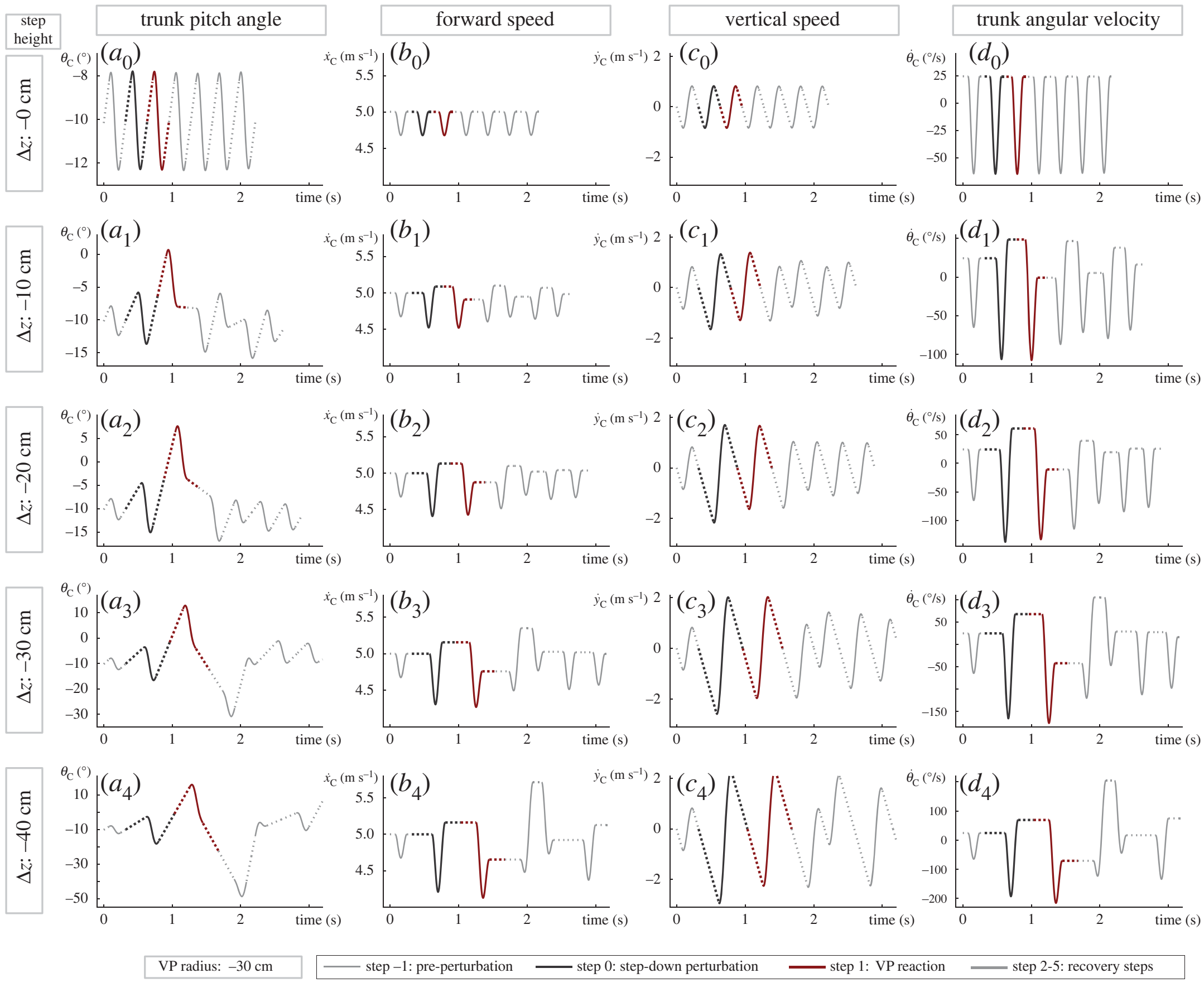


**Figure 13.** The trunk pitch angle (*a*), forward speed (*b*), vertical speed (*c*), and trunk angular velocity (*d*) of the simulated gaits. The figure is complementary to figure 7*a*, which provides the centre of mass trajectories. The solid lines correspond to the stance phase, whereas dotted lines correspond to the flight phase of running in simulation. The step-down perturbation leads to a deviation from the equilibrium state. At the end of the perturbed step (dark grey lines), the forward trunk lean is less, forward speed is higher, and trunk angular velocity is higher than the equilibrium conditions. The step after perturbation (red lines) is the step, where the leg angle and VP controllers react to the changes in state.

## A.4. Simulation: energy regulation at the leg and hip

In figure 14, we present the energy levels of the leg spring, leg damper and the hip actuator for the entire set of step-down perturbations ($\Delta z = [-10, -20, -30, -40\,\text{cm}]$).

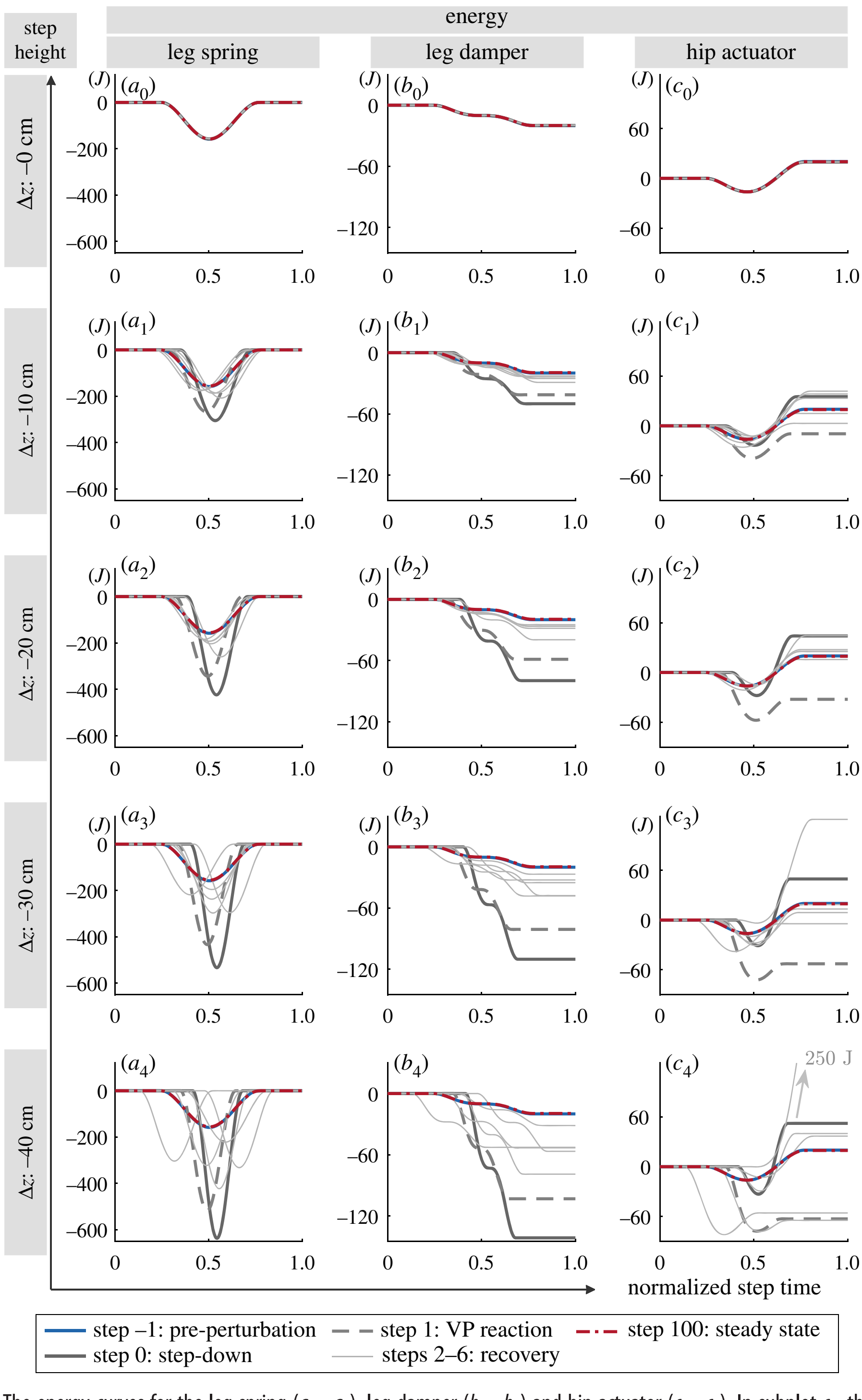


**Figure 14.** The energy curves for the leg spring ($a_0$–$a_4$), leg damper ($b_0$–$b_4$) and hip actuator ($c_0$–$c_4$). In subplot $c_4$, the maximum value of steps 7–11 is indicated with a text and arrow because of the scaling issues. The sub-index '0' indicates the trajectory belongs to the equilibrium state. With the increase of the system's energy at step-down (solid black line), the leg deflects more, the leg damper dissipates more energy and the hip actuator injects more energy compared to its equilibrium condition. During the reaction step (dashed line), the hip actuator reacts to energetic change and starts to remove energy from the system. In the following steps (solid grey line), the hip regulates the energy until the system reaches to the initial equilibrium state (solid blue line). In subplot $c_4$, the maximum value of steps 7–11 is indicated with a text and arrow because of the scaling issues.

## A.5. Simulation: ground reaction forces and impulses

We provide the vertical and horizontal ground reaction forces for the entire set of step-down perturbations ($\Delta z = [-10, -20, -30, -40\,\text{cm}]$) in figures 15 and 16, respectively.

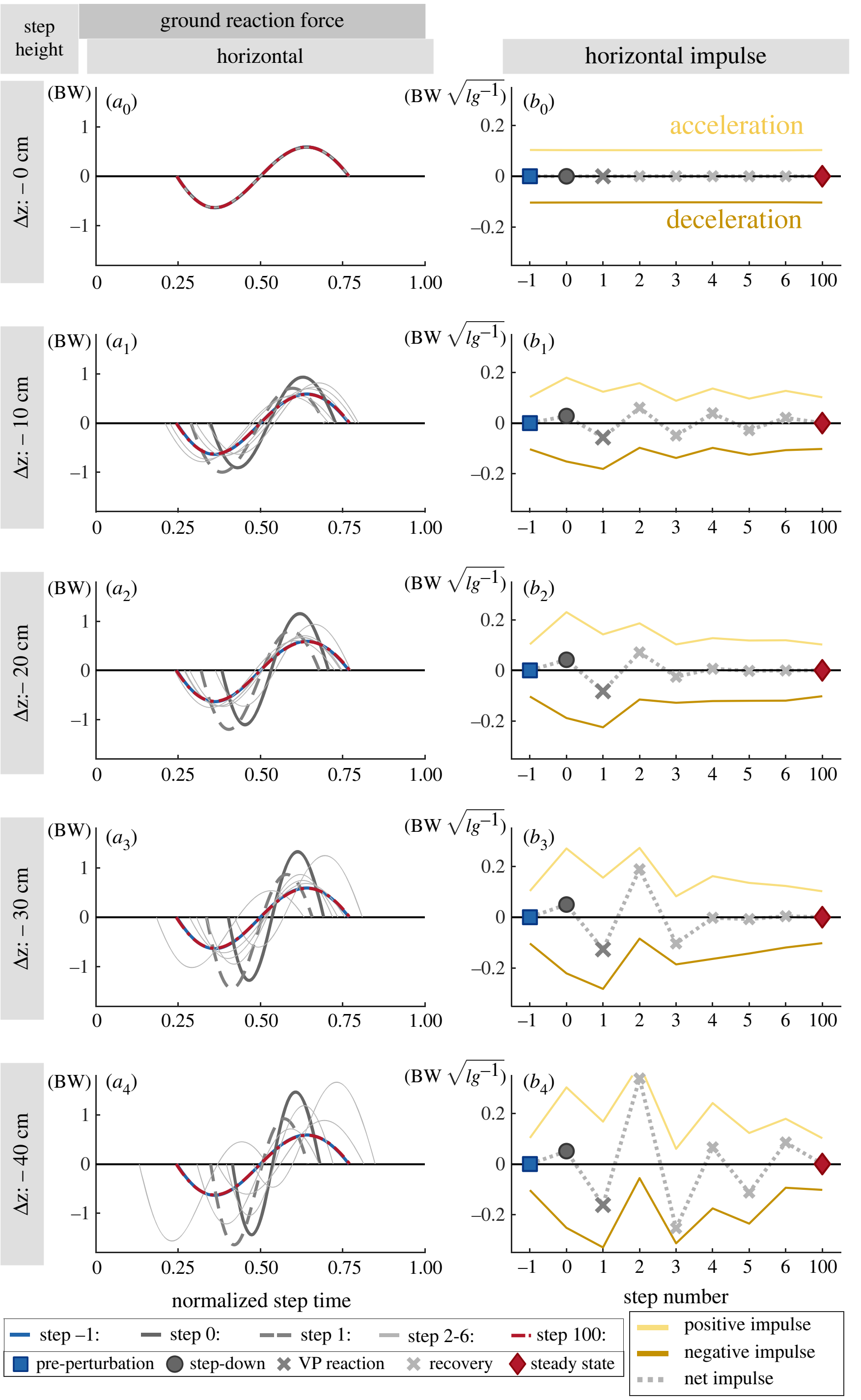


**Figure 15.** The horizontal ground reaction forces over normalized step time are shown ($a_0$–$a_4$). The peak horizontal GRF increases with the step-down perturbation. The area under this curve is the horizontal impulse, which corresponds to the acceleration and deceleration of the main body ($b_0$–$b_4$). The step-down perturbation at step 0 increases the energy of the system. The increase in energy influences the net horizontal impulse, as the impulse attains a positive value (circles) and causes the body to accelerate forwards. In response, the VP position changes to create net negative impulse in the following step (i.e. step 1, dark cross marker) and decelerates the body. The VP position is adjusted until all the excess energy is removed from the system (cross markers) and the gait reaches to an equilibrium state (diamonds).

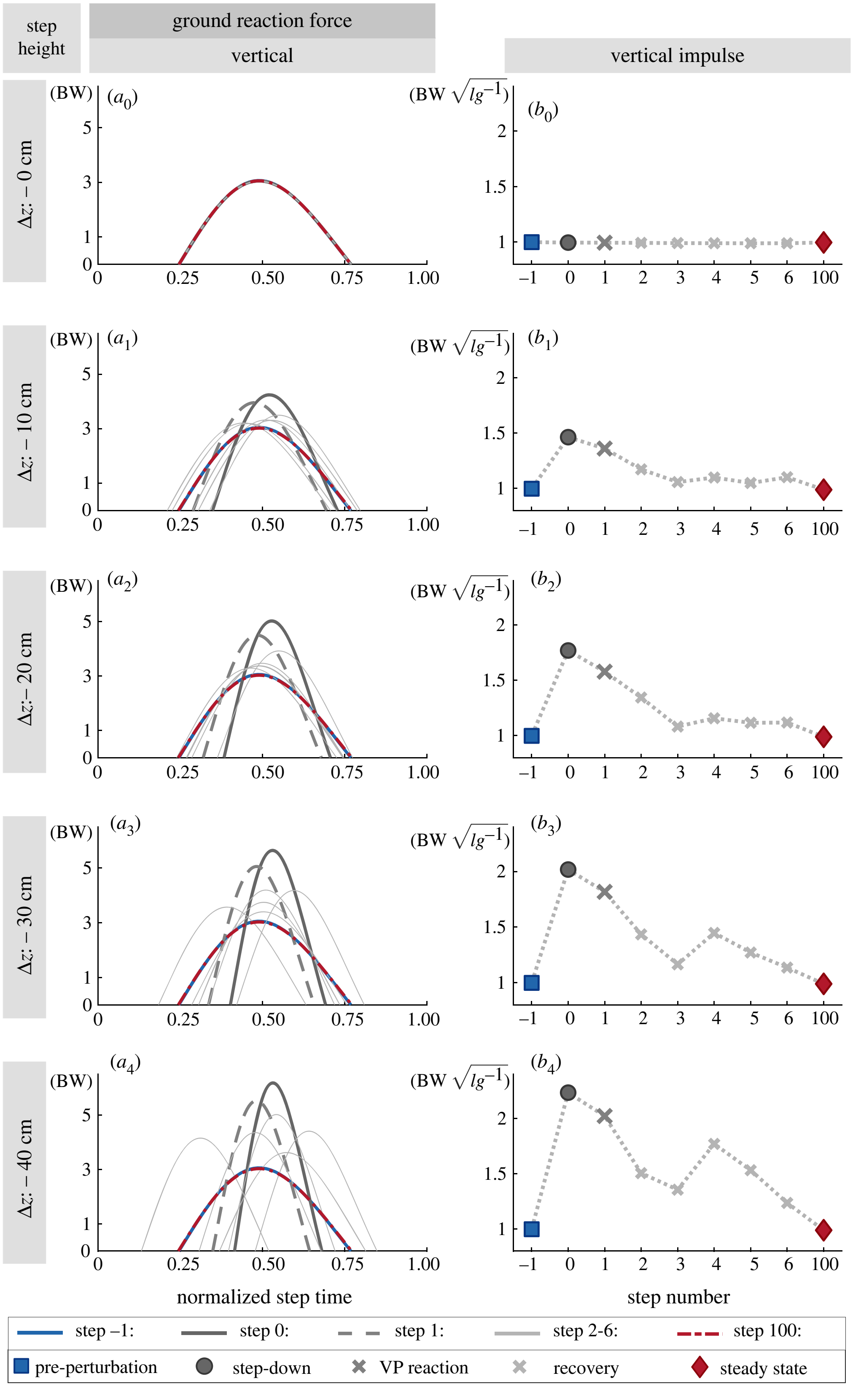


**Figure 16.** The vertical ground reaction forces over normalized step time are shown ($a_0$–$a_4$). The peak vertical GRF increases with the step-down perturbation. During the following steps, the impulse decreases to its initial value through the regulation of the VP position. The increase in the peak GRF after step-down is proportional to the step-down height. Between steps 4 and 5, the peak vertical GRF increases 1.4 fold for −10 cm drop and twofold for −40 cm drop. In accordance, the vertical impulse increases with the step-down perturbation and returns to its initial value ($b_0$–$b_4$). As the step-down height increases from −10 to −40 cm the vertical impulse increases 1.53-fold from its initial value for step 0 (circle) and 1.48 fold for step 1 (dark cross markers).

## A.6. Standard deviation of the experiments

In §4, we provided the standard error (SE) of the measurements from the human running experiments (see the patched areas in figures 9–11). The SE is calculated by dividing the standard deviation (STD) by the square root of number of subjects. The SE shows how good the mean estimate of the measurements is.

On the other hand, STD shows how spread out our different measurements are. The STD is an important measure, especially for the trunk angle measurements, where the trajectories of each subject significantly varies. Therefore, we provide the STD values here for the CoM state in figure 17 and CoM energy in figure 18.

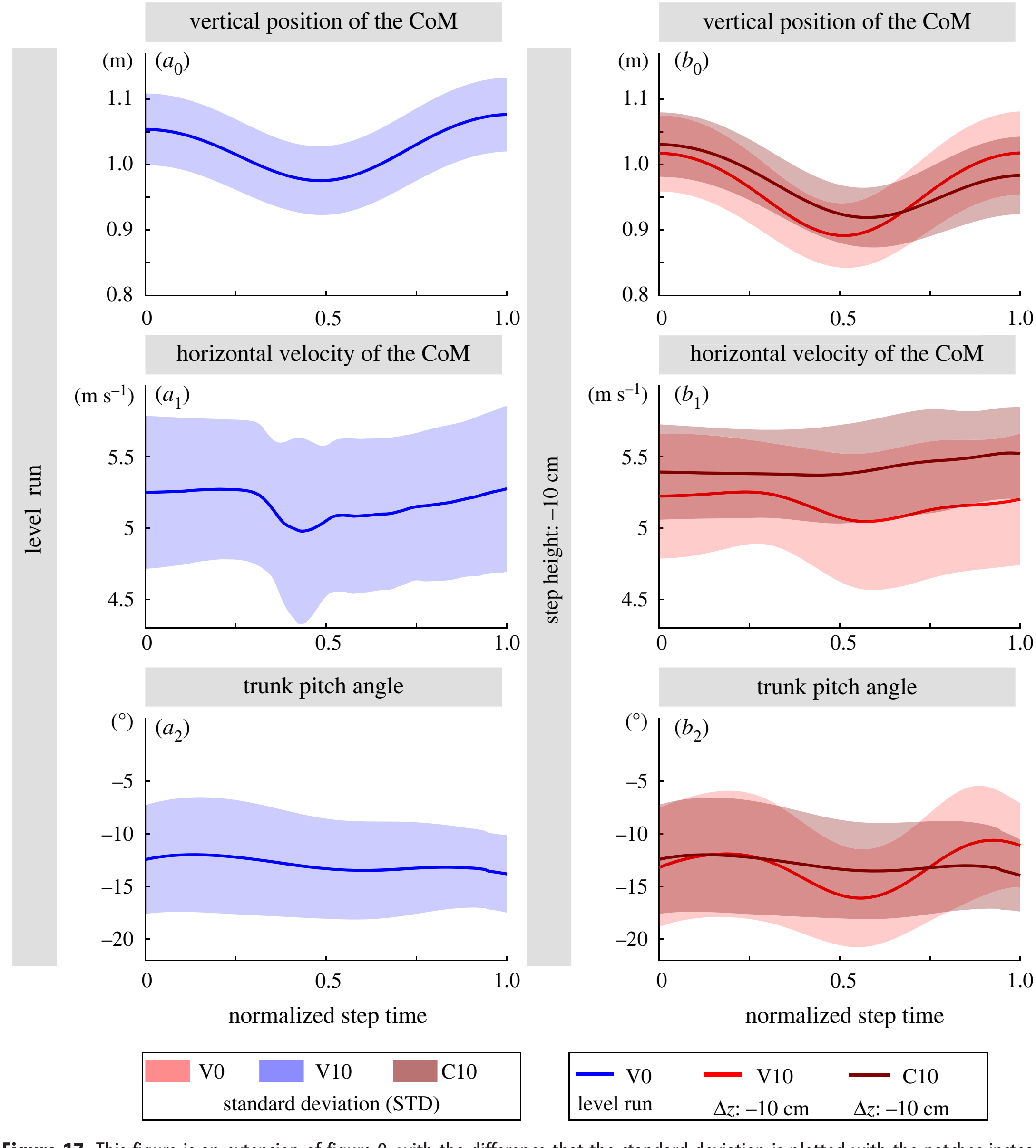


**Figure 17.** This figure is an extension of figure 9, with the difference that the standard deviation is plotted with the patches instead of the standard error.

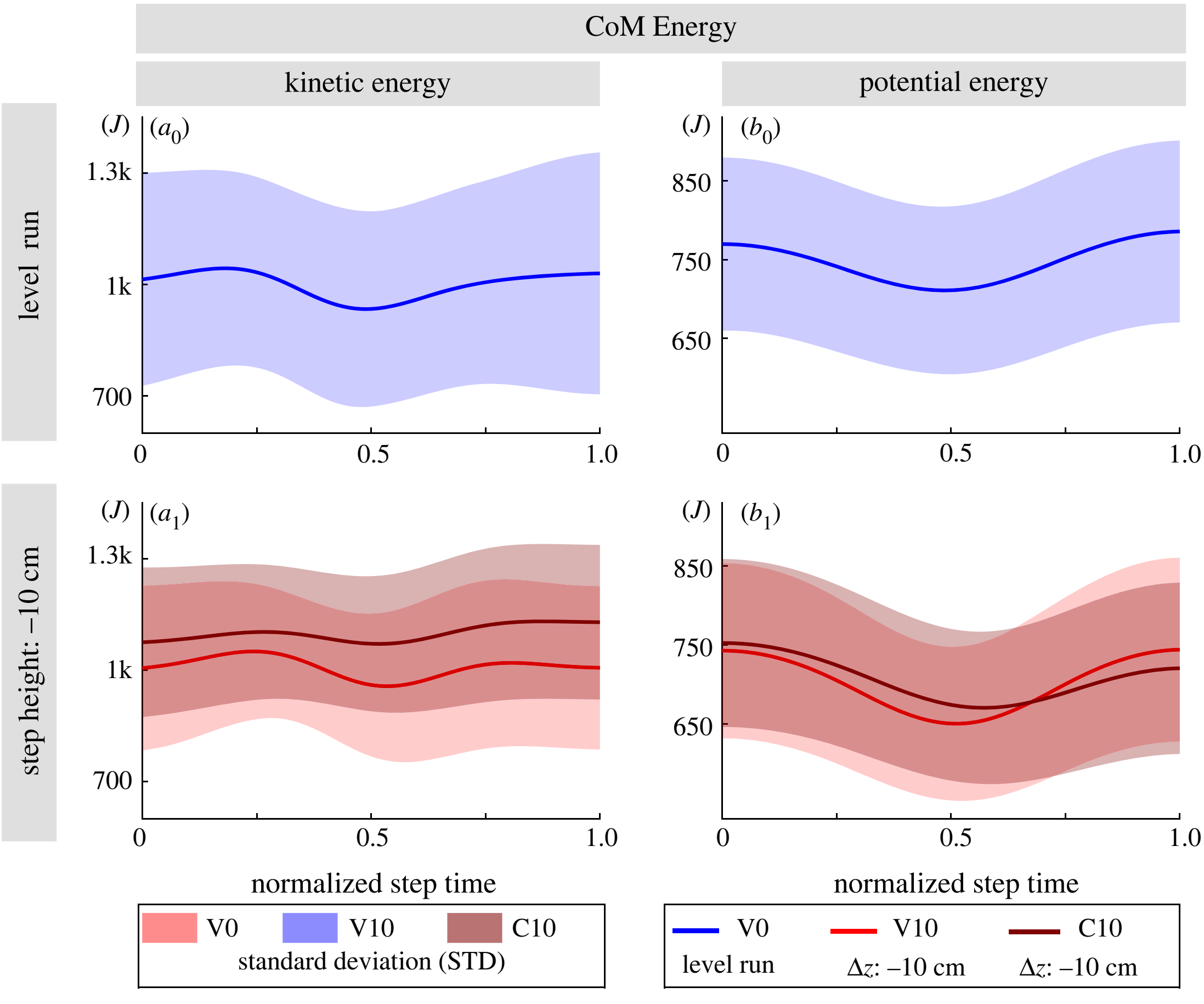


**Figure 18.** This figure is an extension of figures 10 and 11, with the difference that the standard deviation is plotted as patches instead of the standard error.

## A.7. Ground reaction forces: simulation versus experiment

We present the vertical (*a*) and horizontal (*b*) GRFs belonging to the step 0 of the human running experiments (V0, V10, C10) and steps −1, 0 and 1 of the simulations with a −10 cm step-down height, plotted on top of each other in figure 19.

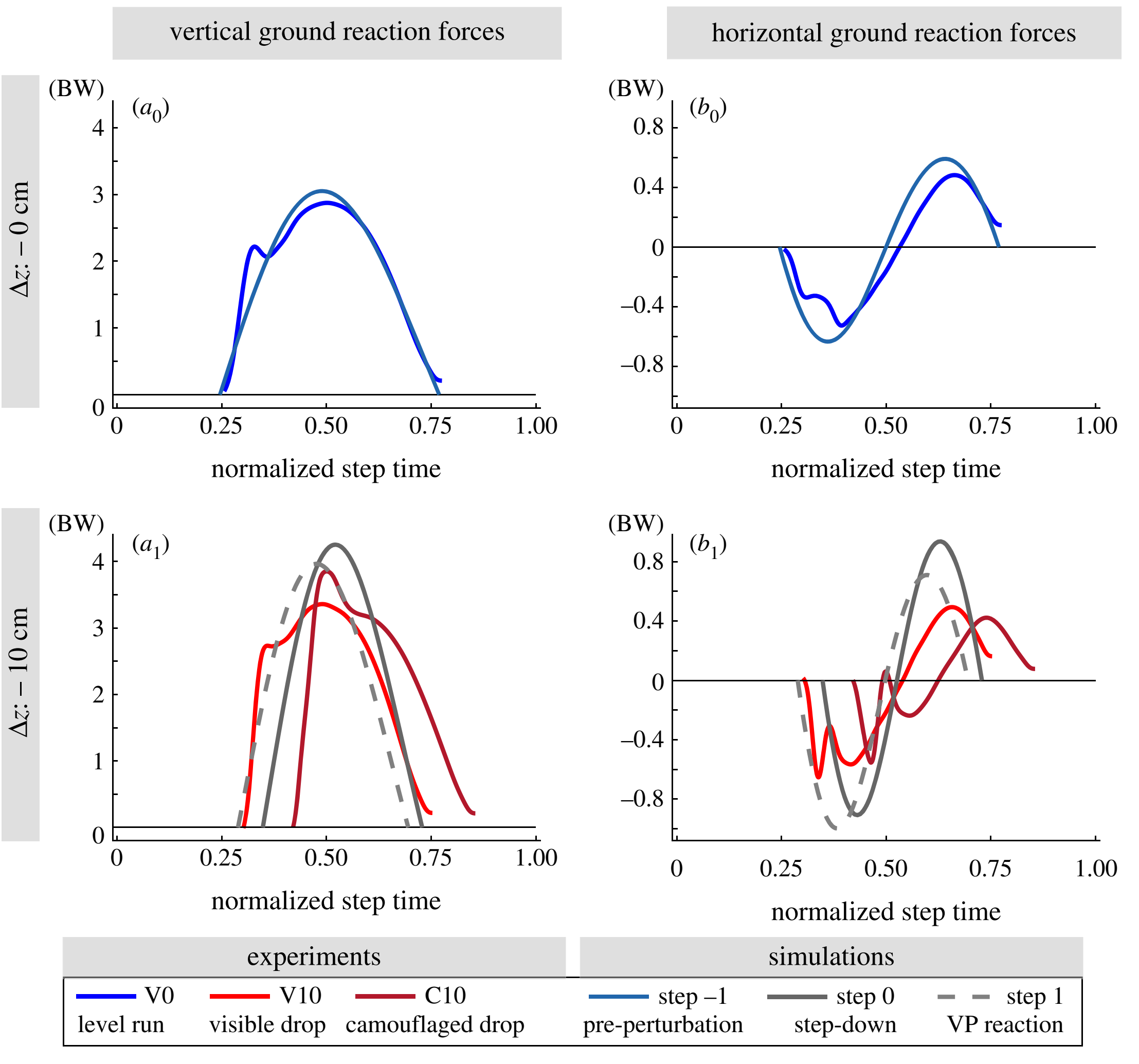


**Figure 19.** The vertical (*a*) and horizontal (*b*) ground reaction forces (GRFs) are plotted over normalized step time. The mean of the experimental results are shown. The TSLIP model simulation is able to capture the characteristics of the GRF in level running ($a_0$, $b_0$). For the step-down perturbation, the model predicts higher values for the peak vertical ($a_1$) and horizontal ($b_1$) GRF, compered to the mean values of the experiments.

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
