## [Reviewer comments · Royal Society Open Science]

Review History

RSOS-200570.R0 (Original submission)

Review form: Reviewer 1

Is the manuscript scientifically sound in its present form?

Yes

Are the interpretations and conclusions justified by the results?

Yes

Is the language acceptable?

Yes

Do you have any ethical concerns with this paper?

No

Have you any concerns about statistical analyses in this paper?

No

Recommendation?

Accept with minor revision (please list in comments)

Comments to the Author(s)

In this study, authors performed experimental and numerical analysis regarding the force direction patterns during ground level running and running on to a visible or camouflaged step-down drop. Experimental results showed that humans tend to generate a VP below the CoM (VPB) for all-terrain conditions and it is backed by the simulation results on the TSLIP model.

Motivation and Novelty - There is limited existing literature that discusses the VPB strategy in humans while walking and no study in case of a running gait. The authors briefly explain the need for studying the VP as a potential way to regulate human's angular momentum during different gait types. The authors anticipate that during the running gait, the trunk should move forward along the pitch axis producing a clockwise angular momentum that would point towards the presence of VPB (which they show through the results). The authors mention further investigation of the ability of a VPB controller to stabilize the gait when perturbed with sudden step-down terrain in the introduction but there is no sufficient explanation or at least it is not coming out well in the discussion between this and the presence of VPB. The latter part becomes important to showcase the novelty of the work as the authors themselves cited various literature on VPB but on different gait and species.

1) Based on results it is presented that for running, VPB is observed, and authors also cited that literature reported VPA. But didn't discuss much on the reason for this difference. It would be better if it is discussed in detail about the reason/ or any difference in assumptions that lead to this etc., that resulted in different VP compared to previous works.

2) After the step down the drop, human has push-off energy, does the controller in TSLIP model account for that? or if not how to include that?

3) It is presented that the simulation results of the TSLIP model are in agreement with the experimental values for level running compared to step-down perturbation. However, the reason is not well discussed.

For instance, in section 4.2, it is presented that using simulation, peak vertical GRFs for step-down perturbations were predicted well but horizontal GRFs are not predicted, reasons for that behavior of the considered model should be discussed (like is from the assumptions/ or some element missing, etc.,)

4) Though authors conducted an experimental and a model-based study and came with observations on identifying VP and implications of VP location on postural stability and energetics, one thing can be improved is creating a proper motivation for the need, experimental observations are used on model to discussed energy distributions, but the model has its own assumptions, like, in running it does not distinguish between the trunk and whole-body dynamics which is not the case in reality. Is it good to extend this to understanding energetics? - some explanation is required on it.

Also, the use of this analysis is to be highlighted, as currently most of the paper discusses the results and inferences, but not on the potential uses. This can be discussed in the introduction and also in results.

5) During the experiment, trials with three conditions were randomized. The rationale for the trials' order is not clear.

Further, a limited number ($n = 10$) of participants were recruited. There is a need to increase the participants' number or proper justification for $n = 10$ is required.

Few minor points:

- The hypothesis needs clarity
- Šidák correction needs some explanation
- Why was the median used for the analysis of the VP position?

- In the last para on page 4, the authors highlight calculations inaccuracy in calculating CoP. What effect would this make on the inferences made or result interpretation?
- In the first para on Page 6, it remains unclear that when VP controller would remain blind to changes in step 0, what would be the effect on the output of the model or the result?
- The fact that there might be a postural adjustment during the step 0 in the experiment, is not discussed in the experimental gait analysis and how would it affect the experimental results obtained. (Page 6)

Decision letter (RSOS-200570.R0)

Dear Ms Drama

On behalf of the Editors, we are pleased to inform you that your Manuscript RSOS-200570 "Postural Stability in Human Running with Step-down Perturbations: An Experimental and Numerical Study" has been accepted for publication in Royal Society Open Science subject to minor revision in accordance with the referees' reports. Please find the referees' comments along with any feedback from the Editors below my signature.

Please submit your revised manuscript and required files (see below) no later than 7 days from today's (ie 06-Oct-2020) date. Note: the ScholarOne system will 'lock' if submission of the revision is attempted 7 or more days after the deadline. If you do not think you will be able to meet this deadline please contact the editorial office immediately.

on behalf of Dr Narayanan Srinivasan (Associate Editor) and Pietro Cicuta (Subject Editor)
openscience@royalsociety.org

Associate Editor Comments to Author (Dr Narayanan Srinivasan):

Associate Editor: 1

Comments to the Author:

One expert reviewer has now commented on the paper. The authors are requested to address all the comments and revise the paper.

Reviewer comments to Author:

Reviewer: 1

Comments to the Author(s)

In this study, authors performed experimental and numerical analysis regarding the force direction patterns during ground level running and running on to a visible or camouflaged step-down drop. Experimental results showed that humans tend to generate a VP below the CoM (VPB) for all-terrain conditions and it is backed by the simulation results on the TSLIP model.

Motivation and Novelty - There is limited existing literature that discusses the VPB strategy in humans while walking and no study in case of a running gait. The authors briefly explain the need for studying the VP as a potential way to regulate human's angular momentum during different gait types. The authors anticipate that during the running gait, the trunk should move forward along the pitch axis producing a clockwise angular momentum that would point towards the presence of VPB (which they show through the results). The authors mention further investigation of the ability of a VPB controller to stabilize the gait when perturbed with sudden step-down terrain in the introduction but there is no sufficient explanation or at least it is not coming out well in the discussion between this and the presence of VPB. The latter part becomes important to showcase the novelty of the work as the authors themselves cited various literature on VPB but on different gait and species.

- 1) Based on results it is presented that for running, VPB is observed, and authors also cited that literature reported VPA. But didn't discuss much on the reason for this difference. It would be better if it is discussed in detail about the reason/ or any difference in assumptions that lead to this etc., that resulted in different VP compared to previous works.
- 2) After the step down the drop, human has push-off energy, does the controller in TSLIP model account for that? or if not how to include that?
- 3) It is presented that the simulation results of the TSLIP model are in agreement with the experimental values for level running compared to step-down perturbation. However, the reason is not well discussed.
For instance, in section 4.2, it is presented that using simulation, peak vertical GRFs for step-down perturbations were predicted well but horizontal GRFs are not predicted, reasons for that behavior of the considered model should be discussed (like is from the assumptions/ or some element missing, etc.,)
- 4) Though authors conducted an experimental and a model-based study and came with observations on identifying VP and implications of VP location on postural stability and energetics, one thing can be improved is creating a proper motivation for the need, experimental observations are used on model to discussed energy distributions, but the model has its own assumptions, like, in running it does not distinguish between the trunk and whole-body dynamics which is not the case in reality. Is it good to extend this to understanding energetics? - some explanation is required on it.

Also, the use of this analysis is to be highlighted, as currently most of the paper discusses the results and inferences, but not on the potential uses. This can be discussed in the introduction and also in results.

5) During the experiment, trials with three conditions were randomized. The rationale for the trials' order is not clear.

Further, a limited number (n = 10) of participants were recruited. There is a need to increase the participants' number or proper justification for n = 10 is required.

Few minor points:

- The hypothesis needs clarity
- Šidák correction needs some explanation
- Why was the median used for the analysis of the VP position?
- In the last para on page 4, the authors highlight calculations inaccuracy in calculating CoP. What effect would this make on the inferences made or result interpretation?
- In the first para on Page 6, it remains unclear that when VP controller would remain blind to changes in step 0, what would be the effect on the output of the model or the result?
- The fact that there might be a postural adjustment during the step 0 in the experiment, is not discussed in the experimental gait analysis and how would it affect the experimental results obtained. (Page 6)

===PREPARING YOUR MANUSCRIPT===

- one version identifying all the changes that have been made (for instance, in coloured highlight, in bold text, or tracked changes);a 'clean' version of the new manuscript that incorporates the changes made, but does not highlight them. This version will be used for typesetting.

===PREPARING YOUR REVISION IN SCHOLARONE===

Author's Response to Decision Letter for (RSOS-200570.R0)

See Appendices A & B.

Decision letter (RSOS-200570.R1)

Dear Ms Drama,

It is a pleasure to accept your manuscript entitled "Postural Stability in Human Running with Step-down Perturbations: An Experimental and Numerical Study" in its current form for publication in Royal Society Open Science.

Best regards,

on behalf of Dr Narayanan Srinivasan (Associate Editor) and Pietro Cicuta (Subject Editor)
openscience@royalsociety.org

Appendix A

Letter to the Editor

We would like to thank the editor and referees for their efforts, time spent, and providing us with such detailed feedback.

Our submission includes the following files:

- 1. root_rev.pdf/.tex:** This document is the revised version of the paper. Changes made are indicated as blue text.
- 2. root_clean.pdf/.tex:** This document is the identical to root_edit.tex, but the changes are not marked with a blue text.
- 3. fig_tab_cap.zip:** This folder includes the pdf files of the figures, xlsx files of the tables, and the txt files of the figure/table captions inside the figs, tabs and caps subfolders respectively.
- 4. RSOS_point_by_point_response.docx:** This document provides point-by-point response to the reviewer.

Appendix B

Point-by-point response to the Comments of Reviewer 1

Summary: In this study, authors performed experimental and numerical analysis regarding the force direction patterns during ground level running and running on to a visible or camouflaged step-down drop. Experimental results showed that humans tend to generate a VP below the CoM (VPB) for all-terrain conditions and it is backed by the simulation results on the TSLIP model.

Response:

We would like to thank the reviewer for the constructive comments and the time spent on the reviewing process. Your feedback is highly detailed and it helped us to improve our manuscript.

Motivation and Novelty: There is limited existing literature that discusses the VPB strategy in humans while walking and no study in case of a running gait. The authors briefly explain the need for studying the VP as a potential way to regulate human's angular momentum during different gait types. The authors anticipate that during the running gait, the trunk should move forward along the pitch axis producing a clockwise angular momentum that would point towards the presence of VPB (which they show through the results). The authors mention further investigation of the ability of a VPB controller to stabilize the gait when perturbed with sudden step-down terrain in the introduction but there is no sufficient explanation or at least it is not coming out well in the discussion between this and the presence of VPB. The latter part becomes important to showcase the novelty of the work as the authors themselves cited various literature on VPB but on different gait and species.

Response:

In our previous simulation work [1], we offered an explanation why VPB can be observed in human level running. We presented a variety of stable gait solutions that trade off between the energy requirements of the leg and hip.

If the VP is an existing function in human gait, it should be present in step-down perturbations as well. Our work here shows that the same principle continues to work in the presence of step-down perturbations. In particular, we show that the VP is able to establish biomechanically similar patterns (i.e., GRF profile and CoM trajectory) for varying terrain conditions (i.e., step-down).

References for Response:

[1] Drama 2019: Trunk Pitch Oscillations for Energy Trade-offs in Bipedal Running Birds and Robots

Comment 1: Based on results it is presented that for running, VPB is observed, and authors also cited that literature reported VPA. But didn't discuss much on the reason for this difference. It would be better if it is discussed in detail about the reason/ or any difference in assumptions that lead to this etc., that resulted in different VP compared to previous works.

Response 1:

The differences came most probably from the different methods analyzing the data. While [1] and [2] only presented single trials of selected subjects, in our study a statistical analysis was performed. Thus, we also document a few outliers that are above the CoM, but without statistical impact. Additionally, [2] most probably omitted the begin and the end of the contact phase and only analyzed the mid part. This can cause a shift in the VPz position.

We added the following part in the Discussion Section 4.1 to make this clear:

Revised text R1: (In Section 4.1 VP quality and location in human gait, page 9)
Previous studies only reported single trials of single subjects and no statistical analysis. We also did observe a few trials as outliers with a VPz above the CoM, which are statistically not significant. Additionally, different ways for cropping the contact phase were considered in the previous studies, which affects the VPz estimation. Here, we consistently crop the first 10% of the contact phase.

References for Response 1:

[1] Blickhan, Andrada, Müller, Rode, and Ogihara 2015: Positioning the hip with respect to the COM: Consequences for leg operation.

[2] Maus 2008: Stabilisierung des Oberkörpers beim Rennen und Gehen.

Comment 2: After the step down the drop, human has push-off energy, does the controller in TSLIP model account for that? or if not how to include that?

Response 2:

We interpret your question in two possible ways: push-off as,

2a. the positive work performed by the ankle extension (plantarflexion) and powered by calf muscles (soleus and gastrocnemius) during late stance phase of the human gait [1],

2b. the mechanical work performed during the propulsion phase of stance phase [2].

2a. Our TSLIP model does not include an ankle joint and an ankle push-off is not considered. However, it is possible to extend the TSLIP model to reflect the ankle push-off by including an ankle torque [3], or push-off impulse [4], or ankle joint [5]. We included this to the discussion briefly.

Revised text R2a: (In Section 4.3 Limitations of this study, page 12)

A future model can be improved by including swing leg dynamics, collision dynamics and ankle torque to capture the heel-strike and ankle push-off effects [3,4,5].

2b. In the following, we consider the push-off energy as the propulsive horizontal GRF energy and look at the horizontal GRF impulse in figure 8b and figure A4 b0-b4 (please refer to the revised manuscript for updated figure numbers).

In our simulation framework, the controller has no dedicated detector for step downs. Such perturbations are captured in the next step with the change in the CoM state. The excess energy is partially dissipated and re-inserted in the next steps through VP

and leg angle controllers (please refer to the explanation in the first paragraph of page 5).

At step 0 (step-down), the push-off energy causes an increase in forward speed (figure A3b), which can also be inferred from the net positive horizontal GRF impulse. In the subsequent transient steps, the leg angle controller tries to reach and maintain the desired forward speed, whereas the VP angle controller tries the same for the desired trunk pitch angle. As a result, the net horizontal GRF impulse fluctuates between positive and negative values, leading to step-wise forward accelerations and decelerations. In the following steps, controllers bring the system to an equilibrium, where the net horizontal GRF impulse and forward acceleration is zero.

We revised the text in Section 3.2.2. to clarify how the push-off energy (inferred from horizontal GRF impulse) causes an increase in forward speed during step-down, and how the leg and VP angle controllers attenuate push-off energy to stabilize the system.

Revised text R2b: (In Section 3.2.2 GRF analysis, page 9)

The net horizontal impulse ... It becomes positive at the step-down perturbation, leading to a net horizontal acceleration. Consequently, the forward speed increases at the end of step 0 (see gray lines in figure A3b). In step 1, the VPBL and leg touch-down angle are adjusted with respect to the change in the state, which leads to a negative net horizontal GRF impulse and decelerates the body (see red lines in figure A3b). In the following transient steps, the leg and VP angle adjustment yield successive net accelerations and decelerations until the system returns to its equilibrium state, where the net horizontal GRF impulse and forward acceleration is zero.

References for Response 2:

- [1] Usherwood 2012: The human foot and heel-sole-toe walking strategy: a mechanism enabling an inverted pendular gait with low isometric muscle force?
- [2] Usherwood 2013: Biomechanics Constraints on muscle performance provide a novel explanation for the scaling of posture in terrestrial animals
- [3] Suzuki and Geyer 2018: A simple bipedal model for studying control of gait termination
- [4] Zamani and Bhounsule 2017: Foot Placement and Ankle Push-off Control for the Orbital Stabilization of Bipedal Robots
- [5] Kim and Collins 2017: Once-per-step control of ankle push-off work improves balance in a three-dimensional simulation of bipedal walking

Comment 3: It is presented that the simulation results of the TSLIP model are in agreement with the experimental values for level running compared to step-down perturbation. However, the reason is not well discussed.

For instance, in section 4.2, it is presented that using simulation, peak vertical GRFs for step-down perturbations were predicted well but horizontal GRFs are not predicted, reasons for that behavior of the considered model should be discussed (like is from the assumptions/ or some element missing, etc..)

Response 3:

3a. We use a simplified model that does not include reflexes, neural control mechanisms, human intent, and sensory feedback mechanisms such as vision, which might play an important role when dealing with step-down perturbations.

3b. Indeed, the differences in the GRFs deserve a more detailed discussion. Thus, we changed the paragraph in the Discussion Section 4.2 and added more details concerning the impact dynamics during the leg touch-down. Another potential reason for the simulation-vs-experiment discrepancy might be the dynamics of the unsprung mass [1] and the impact from heel-strikes. We see in Figure A8.a0-a1 that the impact peak of the vertical GRF is smaller than the active peak for V0 and V10 and is larger for C10. Similarly in Figure A8.b0-b1 the impact peak of the braking GRF is smaller than the active peak for V0 and larger for V10 and C10. In particular, the C10 condition displays a M-shaped braking horizontal GRF, where the impact dynamics become dominant. Our simulation model is not able to capture the impact peak, as it does not include heel-strike. The discrepancy in the peak GRF values might stem from the fact that impact peaks become dominant in step-down perturbations.

Revised text R3b: (In Section 4.2 Experiments vs. model, page 12)

However, differences in the definition of the leg also affect the angle of attack. In the literature, an angle of attack between 57° and 82° was reported for level running [28], which fits the model leg angle. Additionally, no corresponding changes in the braking impulses could be observed. In the perturbed condition the angle of attack in the experiments is 9° steeper compared to V0 because of the swing leg retraction, while the angle of attack in the simulation is the same in V0 and C10. Here, also the braking impulses between experiment and simulation differ. Therefore, we conclude that the simulation could potentially be improved by implementing a swing leg retraction as observed in humans [6, 29, 36].

Another reason for the limited prediction capabilities of the model for step-down drops might be related to the heel strike and impact dynamics during the leg touch-down. The impact peak (i.e., first peak) of the horizontal GRF in the braking interval is smaller than the active peak (i.e., second peak) for V0, and larger for V10 and C10 conditions (figure A8b0-A8b1). In particular, the C10 condition displays a M-shaped horizontal GRF during braking interval, where the impact dynamics becomes dominant. The absence of impact dynamics and associated GRF peaks in the simulation model might contribute to the discrepancy observed in peak GRF magnitudes while stepping down.

References for Response 3:

[1] Schmitt 2011: Human Leg Impact: Energy Dissipation of Wobbling Masses.

Comment 4: Though authors conducted an experimental and a model-based study and came with observations on identifying VP and implications of VP location on postural stability and energetics, one thing can be improved is creating a proper motivation for the need, experimental observations are used on model to discussed energy distributions, but the model has its own assumptions, like, in running it does not distinguish between the trunk and whole-body dynamics which is not the case in reality. Is it good to extend this to understanding energetics? - some explanation is required on it.

Also, the use of this analysis is to be highlighted, as currently most of the paper discusses the results and inferences, but not on the potential uses. This can be discussed in the introduction and also in results.

Response 4:

4a. Our research makes a comparative analysis between human gait data and TSLIP simulation model to identify the commonalities and discrepancies. With the help of our analysis, we can create stronger models in future.

4b. Indeed, there is a discrepancy between the body's and the trunk's CoM. We modified the manuscript to raise this concern, when discussing the discrepancies in trunk pitch angle of the simulation model and experiments. The energetics of our model reflects the body's CoM, and we draw conclusions of the energetics of the leg and hip function.

Revised text R4b: (In Section 4.2: Experiments vs. model, page 10)

Furthermore, the model does not distinguish between the trunk and whole-body dynamics [7].

4c. We included Section 4.4 to discuss the potential uses of VP.

Revised text R4c: (In Section 4.4: Potential uses for the VP, page 12)

Our previous study in [8] offers an explanation why different VP behaviors can be observed in human level running, by suggesting that different VP targets lead to a trade-off between the energy requirements of the leg and hip. In particular, a VP below the CoM (VPBL) with prominent forward trunk motion at ground contact might indicate weaker leg actuation (e.g., caused by injuries), whereas a VPB closer to the CoM or a VPA might indicate weaker hip actuation (e.g., caused by hip extensor strength deficit). Robots and rehabilitation devices could be designed with smaller actuators, after adopting VP positions leading to lower joint loads.

If the VP is an existing function in human gait, the VP based controllers establish biomechanically similar patterns to humans. Consequently, VP based controllers can provide natural reference trajectories for the exoskeletons to assist the human gait with a greater efficiency. For example, VP inspired controllers implemented in the lower limb exoskeleton LOPEZ II [51] and soft passive exosuit in [2] are able to reduce leg muscle activations and decrease the metabolic cost by 10 % and 4%, respectively. Our current work can provide the foundation for a VP-based control approach to assist the human gait in the presence of step-down perturbations.

References for Response 4:

[1] Sharbafi 2013: Robust hopping based on virtual pendulum posture control

[2] de Leva 1996: Adjustments to zatsiorsky-seluyanov's segment inertia parameters.

Comment 5: During the experiment, trials with three conditions were randomized. The rationale for the trials' order is not clear.

Further, a limited number ($n = 10$) of participants were recruited. There is a need to increase the participants' number or proper justification for $n = 10$ is required.

Response 5:

The order of the trials was randomized to exclude the possibility that the trial order influences the outcome through adaption/ learning effects.

Because of the study design (paired tests with repeated measurements), the baseline of the subjects are their own data (level running) and not a control group and the many trials per setting (8 to 15) cause a sufficient effect size. Additionally, the question was whether the VP is above or below the CoM and to answer this question, the effect size was high (Cohen's $D \leq -1.486$). We added the effect size in the Result Section 3.1:

Revised text R5: (In Section 3.1 Experimental Results, page 5)

The VP was in step -1 (pre-perturbed) and step 0 (perturbed) below the CoM ($p \leq 0.001$, Cohen's $D \leq -1.486$) and between -38.8 ± 5.6 cm and -24.0 ± 16.4 cm (figure 4a).

That means, that ten subjects are sufficient to obtain statistical relevant results for the research question. In the existing literature concerning the VP, a similar number of subjects was analysed (e.g. eleven subjects in [1], ten subjects in [2] and ten subjects in [3]).

References for Response 5:

[1] Vilemeyer, Griebach, and Müller 2019: Ground reaction forces intersect above the center of mass even when walking down visible and camouflaged curbs.

[2] Gruben and Boehm 2012: Force direction pattern stabilizes sagittal plane mechanics of human walking.

[3] Müller, Rode, Amniaghdam, Vilemeyer, and Blickhan 2017: Force direction patterns promote whole body stability even in hip-flexed walking, but not upper body stability in human upright walking.

Minor points:

MP.1: The hypothesis needs clarity

MP.2: Šidák correction needs some explanation

MP.3: Why was the median used for the analysis of the VP position?

MP.4: In the last para on page 4, the authors highlight calculations inaccuracy in calculating CoP. What effect would this make on the inferences made or result interpretation?

MP.5: In the first para on Page 6, it remains unclear that when VP controller would remain blind to changes in step 0, what would be the effect on the output of the model or the result?

MP.6: The fact that there might be a postural adjustment during the step 0 in the experiment, is not discussed in the experimental gait analysis and how would it affect the experimental results obtained. (Page 6)

Response to minor points:

R.MP.1a: We rephrased our hypothesis in the introduction as below for clarity.

Revised text R.MP.1a: (In Section 1 Introduction, page 2)

We expect to see a virtual point below the CoM (VPB) shaped by the ground reaction forces, based on the results in [8], and a net forward trunk pitch motion during the stance phase, based on previous results from level running [42]. If the mechanism leading to a VPB in level running remains active, it should also extend to camouflaged, step-down perturbations. Consequently, we hypothesize to observe a VPB also in the step-down experiments.

R.MP.1b: We referred to the hypothesis in the discussion.

Revised text R.MP.1b: (In Section 4.1 VP quality and location in human gait, page 10)
We can also confirm that this point is as hypothesized below the CoM (VPB), as the mean value of the estimated points is -32.2 cm and is significantly below the CoM.

R.MP.2: The Šidák correction is a posthoc test method for the ANOVA and the t-test, respectively. Thus, we expand the manuscript by the following:

Revised text R.MP.2: (In Section 2.1 Experimental methods, page 4)

In order to verify whether the VP is above or below the CoM (VP_A or VP_B), we performed a one-sample t-test compared with zero, separately for each condition with Šidák correction as post hoc test.

R.MP.3: The median was used in order not to weight outliers among the subjects as strongly as the mean value would do.

R.MP.4: Because of the wooden block, the CoP could not be measured directly by the force plates. They can only indirectly measure the pressure of the foot via the wooden block. Because force plate and wooden block are both not idealised flat, there could be inaccuracies in the pressure transfer that could not be quantified. Therefore, it is not possible here to get precise CoP results.

R.MP.5: At step 0, the step-down drop leads to a longer flight phase between apex and touch-down. As a result, the state of the CoM diverges from the equilibrium conditions. The perturbed step ends with a higher forward speed, smaller trunk lean and higher trunk angular velocity (see the figure below). Our previous manuscript included a plot of CoM position and height (figure 7a). We added the figure below to appendix (figure A3) to show the progression of the remaining states.

We rephrased the related text in Section 2.2 for clarity and included the outcome of having a non-reactive perturbation step in Section 3.2.

Revised text R.MP.5.1: (In Section 2.2 Simulation methods, page 5)

During step 0, the state of the CoM diverges from the equilibrium conditions. The postural correction starts at step 1, as the leg touch-down angle and VP angle are adjusted in response to the changes in the CoM apex state.

Revised text R.MP.5.2: (In Section 3.2 Simulation results, page 7)

At step 0, the state of the CoM at leg touch-down diverges from the equilibrium conditions: the trunk pitch angle is smaller (i.e., smaller trunk lean), and vertical speed is higher (see dark gray lines in figures A3a and A3c). The VP position relative to the hip shifts downward, as seen with circle marker in figure 7c1- 7c4. The perturbed state leads to an increase in trunk angular excursion during the stance, whereas the step ends with a higher forward speed, smaller trunk lean, and higher trunk angular velocity (see dark gray lines in figures A3a, A3b, and A3d). At step 1, the leg angle at touch-down is adjusted to a flatter angle and the VP angle to a larger angle (i.e., VP rotates clockwise). The VP position relative to the hip joint shifts to the left, as seen with cross marker in figure 7c1- 7c4. The leftward VPBL shift helps to restore the desired trunk lean and leads to a more pronounced forward trunk motion at step 1 (see red lines in figure A3a).

R.MP.6: Since the postural control of the human step-down running was not the object of the study, we referred to the known literature that considered the postural control. Worth mentioning here is the study by [1], who investigated the swing leg retraction. To make it clearer, we added this example in the Method Section 2.2:

Revised text R.MP.6: (In Section 2.2 Simulation Methods, page 5)

In contrast, small adaptations might already be active at step 0 in the human experiments, e.g., resulting from swing leg retraction dynamics [28, 29].

This aspect of the postural control is then discussed in the Section 4.2, p. 12. Concretely, it is about why the horizontal GRFs are lower in the experiment than in the model which has already been considered in comment 3.

References for Response MP.6:

[1] Müller, Ernst, and Blickhan 2012: Leg adjustments during running across visible and camouflaged incidental changes in ground level.